# Determinants of natural adult sleep: An umbrella review

Nicole Philippens[1]*, Ester Janssen[1], Stef Kremers[1], Rik Crutzen[2]

**1** Department of Health Promotion, NUTRIM School of Nutrition and Translational Research in Metabolism, Maastricht University, Maastricht, The Netherlands, **2** Department of Health Promotion, CAPHRI, Care & Public Health Research Institute, Maastricht University, Maastricht, The Netherlands

* n.philippens@maastrichtuniversity.nl

## Abstract

### Background

Sleep has a major impact on health, which makes it a relevant topic for research and health practitioners. Research on sleep determinants, i.e. factors that positively or negatively influence sleep, is fragmented.

### Objective

The purpose of this umbrella review is to provide an overview of the current evidence on determinants of natural adult sleep.

### Methods

A comprehensive literature search was performed on determinants of sleep. Reviews and meta-analyses on natural adult sleep were included. Six electronic databases (PubMed, WoS, Embase, CINAHL, PsycInfo and Cochrane) were used for the search, last accessed September 2021. The quality of the selected articles was assessed using the AMSTAR2 tool. Results were categorized in four main categories: biological, behavioral, environmental and personal/socio-economical determinants.

### Results

In total 93 reviews and meta-analyses resulted in a total of 30 identified determinants. The impact of each determinant differs per individual and per situation. Each determinant was found to affect different sleep parameters and the relationship with sleep is influenced by both generic and specific moderators.

### Discussion

A comprehensive overview on relevant sleep determinants provides a practical and scientifically based starting point to identify relevant intervention approaches to secure or improve individual sleep quality. The difference in aggregation level of the determinants and in measurement methods are the major limitations of this umbrella review. Extending existing generic sleep hygiene rules with an overview of all types of potential determinants will

**Data Availability Statement:** All relevant data are within the manuscript and its Supporting information files.

**Funding:** The author(s) received no specific funding for this work.

**Competing interests:** The authors have declared that no competing interests exist.

**Abbreviations:** CLI, Combined Lifestyle Intervention; E-type, Evening-type (chronotype); I-type, Intermediate type (chronotype); M-type, Morning-type (chronotype); NAASO, Number of awakenings after sleep onset; NSF, National Sleep Foundation (US); PA, Physical activity; PICO, Population, Intervention, Control and Outcomes; RoB, Risk of Bias; ROL, REM onset latency; SE, Sleep efficiency; SOL, Sleep onset latency; TST, Total sleep time; WASO, Wake after sleep onset.

enhance the awareness of the complexity and can be used to improve the effect of sleep interventions in health promotion.

## Trial registration

The umbrella review was registered with PROSPERO (registration ID CRD42020149648) https://www.google.com/search?client=firefox-b-d&q=CRD42020149648.

## 1. Introduction

Multiple definitions of sleep are available though several core elements of sleep are generally accepted: sleep is a brain process (while the body rests the brain sleeps), sleep is not a unitary phenomenon (it exists of different types of sleep each with their own characteristics, functions and regulatory systems) and some sleep processes are active and involve significant brain activity [1].

Throughout this manuscript we refer to natural sleep as sleep in natural circumstances (e.g. sleep encountered in everyday life) [2]. Determinants are factors that influence health behavior positively or negatively [3]. The concept of determinant in this umbrella review is defined as a factor that influences natural sleep positively or negatively.

Both quality and quantity of sleep are crucial to physical and mental health. Chronic sleep deprivation is related to the development of diabetes and/or obesity. Disturbances of sleep, as indicated by–for example–reduced sleep duration or sleep quality, are causally linked to abnormal glucose metabolism, which might result in diabetes type 2 or metabolic syndrome [4]. Sleep deprivation also affects regulation of appetite and the tendency to skip voluntary physical activity [5] and interacts with self-control, which makes it more difficult to make healthy choices [6]. Chronic sleep deprivation can result in overweight and overweight impacts sleep, making it an even more important topic in addressing overweight and obesity [7–9]. Furthermore, sleep deprivation may also negatively impact emotion regulation, the immune system and job performance and increases the risk of traffic or industrial accidents [10,11]. Regarding work, for example, sleep disturbances are associated with reduced safety and productivity, higher levels of absence from sickness and less career opportunities and job satisfaction [12]. So, the impact of sleep on mental and physical, health-related behavior and quality of (working) life makes sufficient sleep of good quality a corner stone of a healthy lifestyle.

A general rule of thumb for having had sufficient sleep is that an individual wakes up well rested with enough energy to perform well during the day [13]. Longitudinal research shows that sleep duration of 90% of Dutch adults is seven to nine hours, in accordance with the recommendations of the National Sleep Foundation (NSF), that between 11 and 22% experience sleep problems (depending on age and sex; females more than males, elderly more than youngsters), and 45% would like to take action to improve their sleep [14].

A survey by the National Sleep Foundation (NSF) in 2020 reported that 16% of US adults did not feel sleepy during a typical week. Of all the respondents that did report feeling sleepy, 55% indicated not sleeping well enough compared to 44% not having enough time to sleep [15]. In 2013 the NSF compared sleeping habits in six large countries (United States, Canada, Mexico, United Kingdom, Germany and Japan) showing differences between the countries and at the same time insufficient sleep for a significant part. The results to how often the respondents had a good night's sleep differed from 12% (Mexico) to 27% (United Kingdom) [16].

This widely supported need to improve sleep makes it a relevant topic for research and health practitioners. Sleep is a complex phenomenon and typically conceptualized broadly, containing different aspects when operationalized, so called sleep parameters (e.g., sleep duration, sleep quality, sleep disturbances or even some aspects of dreaming). For example, sleep duration is a measure for the length of sleep and sleep onset latency a measure for the time needed to fall asleep after turning off the lights. In the method section definitions of the sleep parameters, that are generally accepted in scientific research, are provided. Different sleep measures shed a different light on experiencing sleep and are subject to different determinants of sleep. As a result, sleep has a broad range of determinants and research is fragmented into different parts of this range, mainly providing in-depth studies focusing on a specific aspect of sleep or sleep regulation. A comprehensive overview of the sleep determinants could act as a solid base for the development of future sleep interventions. Although the relevance of specific sleep determinants varies according to an individual's context and characteristics, the aim of this umbrella review is to provide an overview of the current evidence regarding determinants of natural adult sleep from an individual's perspective. Identification of determinants is an essential step that proceeds selection of most relevant determinants to be targeted in a specific intervention [17].

## 2. Methods

### 2.1 Umbrella review

We have conducted an umbrella review–a meta-review based on previously published reviews and/or meta-analyses. The use of reviews as the source of information provides us with the highest level of scientific evidence on identified determinants. It has the additional benefit of an aggregated level of information, rather than an in-depth study on a single determinant, which fits our aim to provide a high-level overview on all determinants of human adult sleep.

### 2.2 Definitions

The qualification of good sleep contains different parameters of sleep that are covered in the scientific literature. In general, two key aspects of sleep can be identified: sleep duration (i.e. the length of sleep) and sleep quality. Sleep duration and sleep quality need to be regarded as two separate aspects of sleep as correlations between the two might be low [18].

Total sleep length, equivalent to total sleep duration or total sleep time, can be tracked by technological devices (e.g., a wearable) and has a widely supported definition. General consensus has been reached in 2015 on the amount of sleep that we need, i.e. the desirable sleep duration [19]. Sleep quality on the other hand is lacking consensus on a clear definition. In many cases sleep quality is used as an umbrella term to combine several individual sleep parameters, including sleep quantity since both are interrelated as illustrated by the definitions of many sleep parameters. Buysse states that different aspects of sleep quality differ in relevance and importance for each individual [20]. In addition, sleep quality ratings potentially reflect different matters dependent on the individual whereas the term "quality" may be used to define variations in the experience of sleep itself [21]. In 2017, an expert panel initiated by the National Sleep Foundation, reached consensus on the indicators of good sleep quality across the lifespan, thereby identifying and outlining the parameters "sleep latency, number of awakenings, wake after sleep onset and sleep efficiency". No consensus was reached regarding sleep architecture or nap-related variables [22]. All mentioned aspects of sleep quality are considered relevant and, therefore, included in this umbrella review. However, because several reviews did not use these NSF agreed parameters for sleep quality, we had to add some additional

**Table 1. Definitions of sleep parameters.**

| Sleep parameter | Definition | Indication of |
|---|---|---|
| **Total sleep time (TST)** | Total amount of sleep time during a recording period, usually 24 hours (interim wakefulness not included) [23]. | Adequate sleep duration. |
| **Sleep onset latency (SOL)** | Duration of time between the moment the lights are turned off as an attempt to sleep until the time the individual actually falls asleep. [23]. | Bedtime routine, easiness to fall asleep. |
| **Wake after sleep onset (WASO)** | Total of time spent awake after sleep onset [24]. | Reflection of sleep fragmentation, indication of sleep quality and sleep continuity. |
| **(Number of) Awakenings after Sleep Onset (NAASO)** | Total number of awakenings after sleep onset, consisting of short spontaneous arousals (ranging from seconds to minutes) and long-lasting awakenings before the final awakening of a sleep period [25]. | Reflection of sleep fragmentation, indication of sleep quality and sleep continuity. |
| **Sleep efficiency (SE)** | Percentage of total time in bed actually spent asleep [24]. | Overall indication of sleep quality. |
| **REM onset latency[a] (ROL)** | Time from sleep onset until the first episode of Rapid Eye Movement (REM) sleep [23]. | Potential biological marker for sleep related disorders (e.g. apnea) or changes in sleep medication. |
| **Sleep quality (SQ) [a]** | Sleep of good quality generates the feeling to be rested and restored upon waking [26].<br> Symptoms of poor sleep quality: for at least 3 nights per week, having problems on initiating or maintaining sleep, waking up too early or experiencing non-restorative sleep [27]. | General indication of perceived sleep usually combining other sleep parameters such as sleep efficiency, sleep latency, sleep disturbances [28]. |
| **Sleep disturbances (SD) [a]** | Any interruption, measurable or subjectively perceived, from an individual's normal sleep-wake pattern interfering with sleep onset or sleep maintenance [29,30]. | Indication of sleep quality. |
| **Sleep architecture[a]** | Structure of sleep cycles and sleep phases through the night, including five phases i.e. stage 1 (transition to sleep), stage 2 (light sleep), stage 3 and stage 4 (deep sleep) and stage 5 REM-sleep [31]. | Quantification/mapping of sleep characteristics to allow a specific sleep diagnosis and enable comparisons between and intra-individuals. |
| **Insomnia[a]** | Similar symptoms as poor sleep quality, but the symptoms last over a month and lead to consequences in daily life [27]. | Sleep disorder. |

[a]Additional sleep parameters related to sleep quality.

parameters (see Table 1) related to sleep quality. All tracked and self-reported sleep registration was included.

## 2.3 Search strategy

A comprehensive literature search was performed using six electronic databases (PubMed, WoS, Embase, CINAHL, PsycInfo and Cochrane). The search strategy to select the appropriate reviews was set-up in consultation with a senior librarian. The main search terms used for the data extraction were:

- sleep* OR asleep OR time in bed OR bed time OR bedtime OR night rest OR night awak* OR night wak* OR drowsin* OR somnolent* **OR**

- Related MESH terms: "sleep", "sleep wake disorders" and "sleep phase chronotherapy" **AND**

- determinant* OR associat* OR correlat* OR relation* OR relate* OR factor* OR predict* OR influenc* OR effect* OR mechanism* OR parameter* OR impact **AND**

- publication types: reviews, meta-analyses or evaluation studies

No date restrictions or limitations were applied on the searches. Reviews on unpublished studies were excluded from the search result. See S1 File for an overview of all search strings for the different databases.

The searches were re-run prior to the final analysis (in September 2021), in order to identify relevant studies published in the time frame between the first run (in December 2018) and the final delivery of the umbrella review.

## 2.4 Inclusion and exclusion of reviews

The primary inclusion criterion on the population being studied is formulated as: non-hospitalized healthy adults (18+). We included systematic reviews and meta-analyses analyzing determinants of sleep duration and/or sleep quality, written in Dutch, English or German. We studied human sleep, thereby excluding studies on non-humans (e.g. rats, rodents). The focus on healthy natural sleep led to the exclusion of reviews on a specific disease population, illness-oriented reviews on sleep (e.g. sleep and diabetes, asthma, eating disorders, HIV, or AIDS), pain (as signal of an unhealthy condition), sleep diseases (e.g. bruxism, narcolepsy, sleep apnea) or on excess behavior and addictions (e.g. problematic smart phone use, alcoholism). For example a study on the relation between music and sleep was excluded since the populations in this article all suffered from sleep diseases or severe sleep problems [32]. In addition, we excluded articles referring to medical sleep aid devices, pharmacological or herbal treatment including endogenous and exogenous melatonin. As the focus on this study is on adults, we excluded reviews on (unborn) babies, children or adolescents. Since we wanted to identify determinants of natural sleep, we excluded reviews that focus on sleep regulation (e.g. endocrine system), symptoms or consequences of sleep or sleep deprivation (for example sleepiness or diminished cognitive performance due to sleep deprivation) or registration methods of sleep parameters (e.g. actigraphy, EEG, enquiries). For example, a study on sleep and individual differences in intellectual abilities was excluded since it focused on the impact of sleep on intellect instead of vice versa [33]. We excluded studies on sleep interventions (e.g. cognitive and behavioral interventions such as relaxation meditation, controlled breathing and stimulus control). A (behavioral) determinant describes daily routines not necessarily linked to (improve) sleep (which a person normally does) whereas a sleep intervention is specifically targeted at improving sleep (which a person is instructed to do within the context of an intervention). We excluded sleep interventions in line with the objective to study sleep in natural circumstances. We focused on generic determinants of sleep and therefore excluded reviews with such specific target group characteristics that it is no longer a valid representation of the general population (e.g., pregnant women, top athletes). To ensure a basic quality level of the selected reviews we excluded reviews written by one author only, in line with the Cochrane guidelines for systematic reviews [34], and reviews that contained no review question and/or inclusion and exclusion criteria.

Furthermore, we excluded reviews that contained no explicit reference to sleep or sleep-related terminology.

## 2.5 Review approach

We registered the umbrella review with PROSPERO (registration ID CRD42020149648). The database search resulted in a total of 53.512 articles. First of all, the articles retrieved from the search strategy used in different databases were deduplicated between the databases mutually and between the original run and re-run. Next, articles were excluded based on the article title or abstract. The review selection on title and abstract was performed by one author (NP) and checked by means of several random batches by another author (EJ). The remaining articles were fully retrieved (full text) and checked for eligibility. When in doubt articles were discussed by two authors (NP and EJ). Disagreements on inclusion or exclusion at each step were resolved by discussion and when no consensus was reached a third reviewer (RC) helped to resolve the discrepancy. These selection steps resulted in a final set of 93 reviews and meta-analyses that met our criteria to be included in the umbrella review. In Fig 1 a flow chart is presented with each selection step and the number of remaining articles.

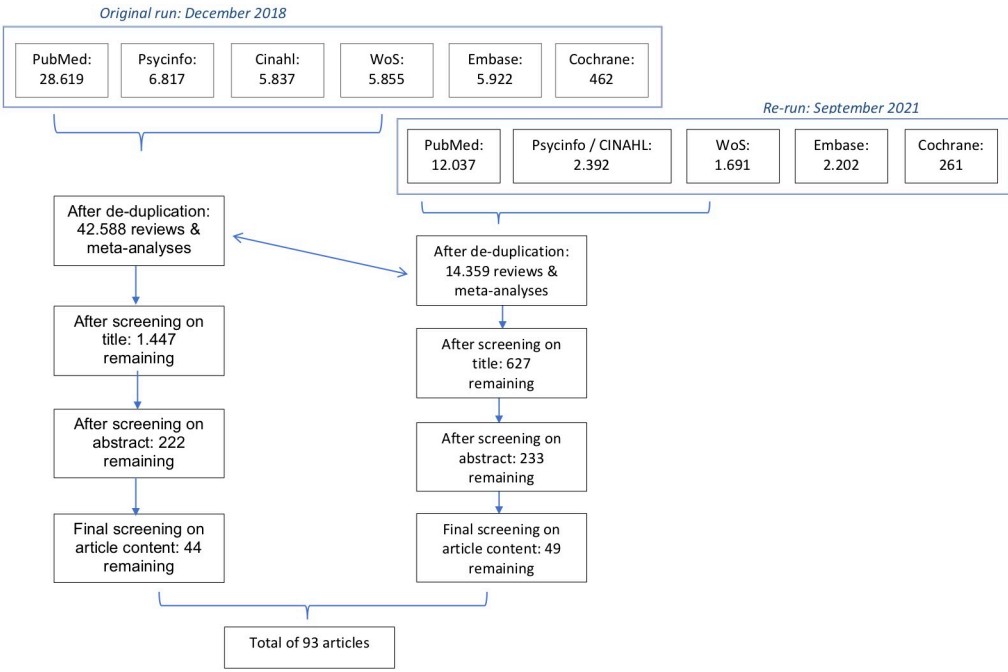

**Fig 1. Flow chart of the article screening process.**

The re-run of the searches resulted in even more articles that met our inclusion criteria despite the much shorter time span of the re-run. This is especially valid for 2021 since the download dates from early October and it is only covering the first three quarters of 2021. This is in line with the number of articles retrieved by our search query on sleep in the last three years as presented in Fig 2.

The data extracted from the selected articles were review characteristics, population studied, number of studies included in the review, identified determinants on sleep in this review, parameters of sleep studied in the review (e.g. sleep duration or sleep disturbances), aspects of sleep affected by determinants (outcomes) and measurement method (e.g. enquiry, actigraphy). See S2 File for an overview of the extracted data.

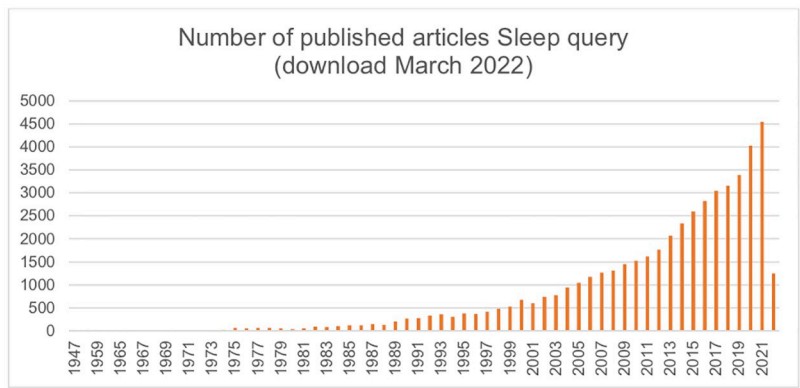

**Fig 2. Published articles in PubMed on sleep.**

**Table 2. Summary of quality scoring of the reviews.**

| Type of review | Quality indication | Parameters |
|---|---|---|
| Meta-analysis | Sufficient quality | Population, Intervention, Control and Outcomes (PICO), search strategy, Risk of Bias (RoB) and heterogeneity explicitly and clearly indicated; an appropriate method for combining results used and RoB and publication bias indicated when discussing results. |
| | Mediocre quality | PICO, search strategy explicitly and clearly indicated; RoB and heterogeneity not or partially defined, appropriate method for combining results used. |
| | Questionable quality | PICO and/or search strategy not explicitly and clearly defined. |
| Review (all other) | Sufficient quality | PICO, search strategy, RoB and heterogeneity explicitly and clearly indicated. |
| | Mediocre quality | PICO, search strategy explicitly and clearly indicated; RoB and heterogenity not or partially defined. |
| | Questionable quality | PICO and/or search strategy not explicitly and clearly defined. |

## 2.6 Risk of bias (study quality)

The quality of the selected articles has been assessed using the AMSTAR2 tool, a validated tool specifically designed to assess the quality of systematic reviews and/or meta-analyses [35]. None of the selected articles scored positive on all critical domains as indicated by the developers of the AMSTAR2 tool. Still there are major differences in the set-up, presentation and discussion of the results between the different reviews, leaving room for a qualitative indication. We therefore used our own qualitative scoring based on several AMSTAR-parameters (Table 2) to distinguish between determinants that are supported by limited scientific evidential value and determinants that are supported by clear evidence. See S3 File for the quality scoring of the selected articles.

A scoping review on the determinants of sleep quality in college students provided a useful starting point for the identification of sleep determinants [36]. We could not use the classification proposed in this scoping review since it covered only part of our (categories of) determinants. Therefore, we clustered and present the sleep determinants in four categories analogous to the classification of the Public Health Classifications Project for Determinants of Health: biological, behavioral, (physical) environmental and personal/socio-economic determinants of sleep [3].

A narrative synthesis of the findings will be provided, structured around the before mentioned categorization of the determinants, the relation with sleep and the specific sleep parameters that are impacted. For practical reasons we combined self-reported and tracked outcomes and used the most commonly used definitions for the sleep parameters. The narrative synthesis is supported by more detailed information presented in Tables 3–6.

## 3. Results

### 3.1 General results

A first general remark on the synthesis is that many reviews identify several generic confounders related to sleep like age, sex, chronotype, BMI or weight, the latter interfering with sleep via hormonal and metabolic pathways and via an increased risk on diseases such as sleep apnea. Generic moderators that were identified are: one's personal attitude towards sleep, medical history, sleep history and baseline sleep, health status (both physiological and mental), economic status, level of education, addictions (including smoking) and (non-sleep) medication.

**Table 3. Overview of biological determinants of sleep.**

| Biological determinants | # reviews | Combined quality indication[b] | Main findings | Determinant specific moderators[c] |
|---|---|---|---|---|
| Age and sex [38,41,42] | 3 | +/- | *Increasing age*:<br>Increase: SD, SOL, WASO, light sleep<br>Decrease: SE, SWS, REM<br>*Women compared to men*:<br>Tracked: higher SQ, longer TST, shorter SOL.<br>Self-reported: lower SQ<br>*Elderly women compared to elderly men*:<br>Elderly men:<br>Increased: variability, light sleep, WASO<br>Elderly women:<br>Increased: deep sleep, NAASO, SOL | Geographic continent |
| Chronotype [39] | 1 | +/- | E-type compared to I-type or M-type:<br>Decrease: TST, SQ, SE (only compared to M-type) | Day of the week (week versus weekend day) |

[b] Clarification of the quality indication combined for all reviews on this determinant: + sufficient quality; +/- mediocre quality;—questionable quality.

[c] Moderators specifically valid for the determinant(s).

On the other hand, several moderators were found to differ per determinant (see Tables 3–6), study set-up and measurement method. Second, some reviews combine several determinants within one of the four categories (biological, behavioral, (physical) environmental or personal/socio-economical) but none of the reviews covers determinants in more than one category.

## 3.2 Biological determinants of sleep

The biological determinants of sleep consist of biological, physiological, somatic (as opposed to psychological), cellular, molecular, organic and genetic affects or characteristics of the body that directly and measurably influence sleep. A review on heritability of sleep showed that as much as 46% of the variability in sleep duration and 44% of the variability in sleep quality is genetically determined [37]. It is practically impossible for an individual to change one's biological determinants of sleep, but they follow the rules of nature and as such can explain sleep differences between individuals and between different life phases. Examples of biological determinants are sex, age and chronotype. In many cases those determinants act as confounders in relation to other determinants of sleep but the factors itself impact sleep independently and interact with each other [26,38–40].

**3.2.1 Age and sex [38,41,42].** With increasing age many sleep parameters show changes: increasing sleep disturbances, sleep latency, wake time after sleep onset and more light sleep. At the same time sleep efficiency, the amount of deep sleep and REM-sleep decreases. Another review found however that increasing age changes sleep timing, towards an earlier bedtime, rather than changes in sleep duration, sleep efficiency or number of awakenings [42].

When measured, women show higher quality of sleep, longer sleep time and shorter time needed to fall asleep than men. However, the self-reported perception of sleep is worse for women compared to men. Combining sex with age: measured sleep parameters show more variability in elderly men than women. The term elderly may refer to different ages but is generalized to ages 58 or 60 and above in the included studies. Comparing elderly men to women: elderly men tend to have more light sleep and elderly women more deep sleep. At the same time elderly women tend to have the highest risk of developing insomnia. Another difference between both is that elderly women have more awakenings at night whereas elderly men have

**Table 4. Overview of behavioral determinants of sleep.**

| Behavioral determinants | # reviews | Combined quality indication[d] | Impact on sleep | Determinant specific moderators |
|---|---|---|---|---|
| Alcohol intake [43] | 1 | +/- | Decrease: SOL, WASO (1st half of night), REM (as % of total)<br>Increase: WASO (2nd half of night), SWS (1st half) | Alcohol dosage<br>Timing of alcohol intake |
| Caffeine intake [44] | 1 | +/- | Decrease: TST, SE, SWS, SQ<br>Increase: SOL, NAASO | Caffeine sensitivity<br>Efficiency of caffeine metabolism<br>Timing of intake |
| Intermittent fasting [45–47] | 3 | +/- | Decrease: TST, REM<br>Shift bedtime and risetime during Ramadan | Delayed bedtime<br>Meal composition<br>Caloric intake<br>Energy expenditure |
| Diet [48–52] | 5 | +/- | Insufficient evidence for causal relation with sleep.<br>Dietary components increasing SQ: Tryptophan-rich food, cherries or cherry juice, milk, well-balanced diet, healthy diet[e] | Meal composition |
| Micronutrient intake [48,49,53] | 3 | +/- | Sufficient intake supports SQ.<br>Especially valid for zinc, vitamines D and E (elderly)<br>Effect of nutrients ambiguous on:<br>TST, SOL, WASO, SWS | |
| Macronutrient intake [54,55] | 2 | +/- | **Acute intake**:<br>Carbohydrates no effect<br>Protein/fat: inconclusive<br>**Long-term intake**:<br>High carbohydrates: increase REM<br>Low carbohydrates: increase SWS<br>Excess protein: decrease SQ<br>Fat: inconclusive<br>**Energy restriction for overweight and obese individuals**:<br>Higher protein: increase SQ<br>Carbohydrates/Fat: inconclusive | Daily energy intake<br>Glycemic load<br>Meal composition |
| Physical activity [23,28,57–70] | 16 | + | **PA in general** beneficial effect on one or more sleep parameter(s):<br>Increase: TST, SQ, SE<br>Decrease: SOL<br>**Longer duration PA**:<br>Increase: TST, SWS<br>Decrease: SOL, REM<br>**Other PA parameters** (intensity and mode): no significant differences<br>**Evening exercise**:<br>No detrimental effect, except for<br>vigorous[f] exercise less than 1 hour before sleep: increase SOL, decrease TST | Perceived intensity of exercise<br>Fitness level<br>Mood regulation, anxiety, arousal<br>Psychological functioning<br>Mode of exercise<br>Duration of exercise<br>Timing of exercise<br>Core body temperature at bedtime<br>Stress level |
| Meditational physical activities [71–75] | 5 | + | Increase: SQ | Geographical location<br>Frequency, duration and intensity of exercise |
| Sedentary behavior [63,76,77] | 1 | + | **Sedentary behavior**:<br>Increase: SD, risk of insomnia | Physical activity level |
| Gaming and social media [63,76,77] | 2 | +/- | **Gaming**:<br>Increase: SOL<br>Decrease: SQ, TST<br>**Social media (pre bedtime)**:<br>Decrease: SQ | Violence level of game<br>Exposure time<br>Duration of game time<br>Desensitizing effect in players<br>Location of gaming/social media use (bedroom, elsewhere) |
| Cognitive activity [78] | 1 | +/- | Decrease: WASO, NAASO<br>Other sleep parameters no effect. | |
| Music listening [79] | 1 | +/- | Increase: SQ (self-reported) | |

[d] Clarification of the quality indication combined for all reviews on this determinant: + sufficient quality; +/- mediocre quality;—questionable quality.

[e] Healthy diet: rich in plant-based foods and seafood together with low processed food and low sugar-rich food.

[f] Vigorous is defined at an intensity >76% HR-max or >69% VO2-max.

**Table 5. Overview of (physical) environmental determinants of sleep.**

| Environmental determinants | # reviews | Combined quality indication[g] | Impact on sleep | Determinant specific moderators |
|---|---|---|---|---|
| Disasters [80,81] | 2 | +/- | Rising temperature: decrease: TST, SQ<br>Extreme weather, floods & wildfires: decrease SQ<br>COVID-19: weak association with sleep problems (insomnia) | Income<br>Fear |
| Air quality [82–84] | 3 | +/- | **Distinctive scents:**<br>Increase: SOL<br>Decrease: SQ<br>**Decreased oxygen:**<br>Decrease: SE, SQ (self-reported)<br>**Air pollution:**<br>Decrease: SQ | Season<br>Humidity<br>Temperature |
| Ambient temperature [82,85] | 2 | +/- | **Within thermal neutral range:**<br>Cooler: increase SWS, SQ (men)<br>Warmer: decrease SOL<br>**Above/below thermo neutral**<br>Increase: NAASO/SD<br>Decrease: SQ | Bedding and clothing<br>Humidity<br>Ventilation/airflow<br>Body and skin temperature |
| Noise [29,30,86–91] | 8 | +/- | **Traffic (road/rail/air):**<br>Decrease: SWS (air)<br>Increase: SD, WASO<br>**Windturbines:**<br>Increase: SQ (self-reported)<br>Ambiguous on tracked sleep parameters<br>**Ambient noise:**<br>Meaningful: increase SD<br>Non-meaningful: ambiguous outcomes<br>Low-frequency: dose-response self-reported sleep quality | Noise sensitivity<br>Neuroticism<br>Extraversion<br>Annoyance<br>Duration, loudness, timing of noise<br>Continuous/intermittent<br>Base level background noise<br>Wind speed and direction<br>Topography<br>Economic aspects<br>**Windturbines:**<br>Visual interference of turbines |
| Light [82,92–95] | 5 | +/- | **Light in general:**<br>Positive effect on sleep<br>**Bright (>1000 lux) day light:**<br>Increase: SQ (self-reported)<br>**(Bright) light at night:**<br>Decrease: SQ<br>(dose response with brightness and duration) | Intensity of light<br>Timing of light<br>Duration<br>Color spectrum/wavelength<br>Prior exposure to light<br>Continuous/interrupted<br>Indoor/outdoor<br>History of light<br>Rate of change in light |
| Green space [96] | 1 | + | Decreased risk on short TST or insufficient SQ | Week or weekendday |

[g] Clarification of the quality indication combined for all reviews on this determinant: + sufficient quality; +/- mediocre quality;—questionable quality.

longer awakenings. Elderly men spend more time in bed and elderly women need more time to fall asleep once they are in bed.

**3.2.2. Chronotype [39].** Chronotype is the natural inclination of an individual about when they prefer to sleep and when they prefer to stay awake. Chronotype results from the underlying biological circadian rhythm. Three different chronotypes can be distinguished, e.g. M-type (morningness), I-type (intermediate) and E-type (eveningness). Approximately 80% of the human population matches the I-type whereas M-type and E-type each cover 10% of the population. E-type, compared to I-type or M-type, shows more often a decrease in sleep time, decreased subjective sleep quality and less sleep efficiency (the latter when compared to M-type only). E-type in addition show more irregularity in their sleep schedule.

**Table 6. Overview of personal and socio-economic determinants of sleep.**

| Personal and socio-economic determinants | # reviews | Combined quality indication[h] | Main findings | Determinant specific moderators |
|---|---|---|---|---|
| Attachment style [97] | 1 | - | High anxiety: decrease SQ, effect stronger when combined with high avoidance | Depression Quality of current relationship General level of anxiety Life satisfaction |
| Sexual orientation [98] | 1 | +/- | Sexual minorities: Decrease: TST, SQ | Social context |
| Psychological dispositions [26,99–102] | 5 | + | **Negative affect**: no significant effect **Positive affect**: increase SQ (self-reported) **Humor and laughter**: no significant effect **Self-compassion**: increase SQ **Self-coldness**: decrease: SQ (stronger association) **Worry and rumination** Decrease: SQ, TST Increase: SOL | Family (children) Bed partner Psychological distress Social norms |
| Ethnicity [103–105] | 3 | +/- | **African versus Caucasian American**: Longer SOL, more light sleep, less SQ, less SWS, less SE. Effect stronger for women, disappears with increasing age. **Minorities**: Shorter TST, less SQ | Marital status Stress experience Reappraisal of health status Employment status Discrimination Sub ethnicity Urbanization |
| Work [27,106–112] | 9 | +/- | **On-call**: decrease TST, other parameters ambiguous. **Job demands (higher)**: decrease SQ **Job control**: ambiguous **Positive support**: Co-workers: no effect From leaders: decrease: SOL, SD **Negative support**: Increase: SD Decrease: TST (work stressors), SQ **Bullying at work**: Increase: sleep problems Decrease: TST **Violence at work**: Decrease: SQ Increase: SOL, WASO **Shiftwork** Morning shifts: decrease TST Evening shifts: increase TST Night shifts: contradictory on TST, SE no effect Increase: SOL, WASO. Rapid versus slow rotation: Increase: SD Decrease: TST | Workload Fatigue Positive and negative affect Work-family conflicts Physical activity (in leisure time) Relaxation Employment type Detachment from work Shift rotation schedule (permanent, rapid, slow) Habituation to shiftwork |
| Psycho-social stress [114–116] | 2 | +/- | Daily life stress Increase: WASO Other parameters: inconsistent Stressful life events (bereavement) Per stressor/individual different parameters impacted. Increase: REM, WASO Decrease: ROL, SWS Experimental stress Increase: WASO, SOL Decrease: SWS, REM, SE Discrimination: poorer sleep | Coping ability Type of stressor (institutional, race-related, daily life, life events) Chronicity (duration) Virulence Diversity of life (stress) Chronic pain |

(*Continued*)

**Table 6.** (Continued)

| Personal and socio-economic determinants | # reviews | Combined quality indication[h] | Main findings | Determinant specific moderators |
|---|---|---|---|---|
| Social relations [117–124] | 8 | +/- | When co-sleeping and/or good relationship quality: Increase: SWS, SQ Decrease: SOL, REM Parent-child (increased SD for child): Decrease: SQ, TST, SE Increase: NAASO Social participation: Increase: SQ (women) Social support from partner: Increase SQ No effect on TST or SE. Positive romantic relation: Increase SQ (tracked and self-reported) Decrease TST Negative relation (abusive, aggressive): Decrease SQ Increase: SD Loneliness (effect stronger for males) Increase: SD, risk of insomnia Decrease: SQ | Cultural beliefs Chronotype "match" of bedpartners Sleeping disorder of bedpartner Life events Personality traits Household Family practices and interactions Type of social support Emotion and anxiety regulation Positive affect Interpersonal stress |
| Socio-economic status [126] | 1 | +/- | Lower SES: Decrease: TST, SQ Higher SES/higher income/higher education: Decrease: SOL Increase: SE, TST Difficult socio-economic position: Increase: WASO | |
| Seasonal and cultural patterns [127] | 1 | + | No significant effect on sleep (TST). | Geographic location Other behavior (sedentary / physical activity) |

[h] Clarification of the quality indication combined for all reviews on this determinant: + sufficient quality; +/- mediocre quality;—questionable quality.

Table 3 provides an overview of all identified biological determinants of sleep.

## 3.3 Behavioral determinants of sleep

Behavioral determinants consist of activities, actions or patterns of actions undertaken by individuals with the potential to influence sleep. More specifically, behavioral determinants include risk and protective behaviors related to sleep.

**3.3.1 Alcohol [43].** The effect of alcohol on sleep can be split in two distinctive areas of impact: the first half and the second half of sleep. Alcohol in all doses (from one drink per night to over four drinks) leads to less time needed to fall asleep. It also leads to less disruption during the first half of sleep and more sleep disruption in the second half of sleep. It increases the percentage of deep sleep during the night proportional to the amount of alcohol intake: more alcohol means more deep sleep. On the other hand, the higher the dose, the more sleep disruption (WASO) occur and the lower the percentage of REM-sleep.

**3.3.2 Caffeine intake [44].** The review on caffeine is based on studies on male adults in Western countries. There is a clear relation between caffeine intake and sleep. Caffeine shows a negative effect on total sleep time, a prolonged effect on sleep latency, more awakenings during sleep, reduced sleep efficiency and reduced deep sleep. The perceived self-reported sleep quality is also negatively affected though in general the perception of the effect of caffeine on sleep is underestimated.

**3.3.3 Intermittent fasting [45–47].** The reviews on intermittent fasting are focused on Ramadan studies in Arabic countries. During Ramadan, with a timespan of approximately one month, adults refrain from food or fluid intake from sunrise to sunset. The studies show an effect on sleep-wake patterns, a decrease in sleep duration as well as REM-sleep but these effects might be directly linked to the early morning rise during Ramadan instead of intermittent fasting itself. In addition, the influences on sleep might reflect other behavioral lifestyle changes, beyond fasting and changed bedtimes, during Ramadan.

**3.3.4 Diet [48–52].** In general, there is insufficient evidence for dietary interventions to modulate sleep. Conclusions regarding single nutrients are difficult to draw due to the number of nutrients consumed together. Causal relationships between nutrients and sleep are not yet available. However, some diets and nutrients look promising in supporting sleep of good quality: healthy diets (i.e. rich in plant-based foods and seafood together with low processed food and low sugar-rich food), well-balanced diets (including milk and dairy products), specific nutrients (tryptophan) as well as diets containing products with high concentrations of melatonin such as eggs, milk, kiwis and sour cherries. Whereas most studies show that a diet rich of these products facilitate sleep of good quality, for older adults a similar but negative relation has been found: less milk and less fish may have a negative impact on sleep.

**3.3.5 Micronutrients intake [48,49,53].** The studies included three types of micronutrients: vitamins, trace elements and minerals. In general, it can be concluded that an optimal dose of micronutrients is needed to maintain healthy/normal sleep especially regarding zinc for the general population and vitamin D and E for older adults. No unambiguous effect of a single micronutrient could be identified. Most studies focused on short-term effects whereas deficiencies in micronutrients slowly build up and might have an impact on sleep across the lifespan.

**3.3.6 Macronutrients intake [54,55].** Acute manipulation of carbohydrates is unlikely to affect sleep whereas long term manipulation shows that higher carbohydrates intake prolongs REM-sleep at the cost of deep sleep, whereas lower carbohydrates intake shows the reversed outcome. Both quantity and quality of carbohydrates intake does not affect any other sleep parameter. The outcomes on protein and fat are inconclusive except for protein intake exceeding acceptable limits: it will have a detrimental effect on sleep. For overweight and obese individuals on energy restriction, higher protein intake may positively affect sleep. However, there is a strong association between weight loss and sleep duration and sleep quality which may confound the outcomes when manipulating macronutrient intake [56].

**3.3.7 Physical activity (PA) [23,28,57–70].** The definition of PA differs per review. We used the broadest definition that was used in these reviews: "any form of body movement that results in energy expenditure above basal level" [62]. An umbrella review of PA and sleep on all reviews published between 2006 and 2020 showed strong evidence that both acute and regular sessions of PA improve sleep outcomes among adults [68]. This umbrella review combined many different forms of PA, varying from walking and stretching to intense aerobic activities such as running, and various target groups, varying from athletes, menopausal women and adolescents to elderly. For sessions of longer duration, independent of intensity, there is moderate evidence for sleep changes, i.e. an increase in sleep duration and deep sleep and a decrease in sleep latency and REM-sleep. With increasing age, the advantage of decreased sleep latency diminishes.

Single reviews show other insights into the relationship between PA and sleep though the results are mixed. In general, PA has a slight positive effect on self-reported sleep quality; moderate activity seems to improve sleep quality whereas the role of vigorous activity in influencing sleep needs more research.

High intensity evening exercise does not disrupt nighttime sleep of healthy adults, except when this is done close to bedtime. Sleep and PA might be considered as dependent behaviors rather than exposure (PA) and outcome (sleep): both activities compete for time within the 24-hour daily timespan.

**3.3.8 Meditational physical activity [71–75].** Meditational physical activities can be defined as a form of exercise focusing on strength, flexibility and breathing to boost physical mental and spiritual health. These exercise forms (e.g. pilates, yoga, tai-chi or qigong) can be both vigorous or more relaxed and mindful. The reviews on this topic, mostly done on older people living in Asia, are based on a limited set of articles, but the reviews are of sufficient quality. The outcomes on sleep are mixed with a strong tendency towards favorable outcomes on sleep quality. The reviews are contradictory on the effect for different populations: some show the largest effect for healthy adults, others for population with sleep problems. One of the reviews identified the geographic location as a strong moderator for the effect of meditational physical activities (i.e. Asian populations showed a large effect size whereas the US populations showed no effect) [74].

**3.3.9 Sedentary behavior [63,76,77].** Sedentary behavior, characterized by a sitting or reclining posture and low-energy expenditure, is practically the opposite of being physically active while both behaviors generally alternate during human daily life. Whereas physical activity in general positively affects sleep, sedentary behavior increases sleep disturbances and may increase the risk of insomnia.

**3.3.10 (Video) Gaming and social media [63,76,77].** Video gaming in general shows an increase in sleep latency and a decrease in sleep quality and total sleep time with the side mark that there are several influential moderators such as the duration of gaming and violence level of the game.

The use of social media, defined as all tools to communicate between networks, internet and mobile phones, is far greater among young people than older generations. Social media use prior to bedtime can result in lower sleep quality. Social media acts as a major source of information and receiving incorrect information on sleep might even pose a greater threat making health literacy (on the topic of sleep) an important mediator.

**3.3.11 Cognitive activity [78].** Studies on sleep following intense cognitive activity, mainly intensive learning tasks, show mixed results. Most studies report no changes on sleep parameters whereas few studies report positive changes in sleep. The most prominent changes are a decrease in wake after sleep onset and less arousals from sleep.

**3.3.12 Music listening [79].** Listening to music prior to bedtime for 30 to 60 minutes, regardless of musical genre, improves sleep quality. The positive effect diminishes over time but has no deteriorating effect on sleep. The music seems to replace bad habits (rumination and hyperarousal) and is most effective for individuals with mild sleep problems.

Table 4 provides an overview of all identified behavioral determinants of sleep.

## 3.4 (Physical) Environmental determinants of sleep

(Physical) environmental determinants exist of any external factor (biological, chemical, physical) that can be linked to a change in sleep. Environmental determinants include the characteristics of the sleeping environment, hazards and effects of climate and geography. Social and cultural aspects of the environment are included in the category of personal/socio-economic determinants.

**3.4.1 Disasters [80,81].** Climate change can impact sleep via inherent determinants (e.g. rising temperature, floods and fires and extreme weather events). All these factors impact sleep quality whereas a rising environmental temperature has the most direct impact on sleep quality

and also leads to a decrease in sleep duration. Some important side notes: the impact of catastrophic events might act as major stressors and indirectly impact sleep. Recently a new worldwide stressor has evolved: COVID-19. Extreme fear for example for social isolation, uncertainty, illness and/or financial problems, is the most prevalent response to pandemic diseases. The review on fear of COVID-19 and sleep problems (insomnia) showed a weak correlation [81].

**3.4.2 Air quality [82–84].**   Distinctive scents in the direct environment, like room scents, smoking or cooking fumes, can decrease the quality of sleep and increase sleep latency. A decreased oxygen level in the sleeping environment, e.g. when sleeping on altitudes over 2000 meters or in an unventilated bedroom, may decrease sleep efficiency and may generate decreased perceived sleep quality.

Ambient air pollution is a mixture of particulate matters (PM) and gaseous components evolving from harmful natural, human or industrial sources such as smoking, vehicle emissions and factory outputs. Air pollution has an adverse effect on sleep but above all air pollution increases the prevalence of respiratory problems related to sleep such as sleep disordered breathing.

**3.4.3 Ambient temperature (and humidity) [82,85].**   When considering the impact of ambient temperature on sleep, body and skin temperature regulation, airflow, humidity, clothing and bedding should also be taken into account. The experienced temperature is closely related to humidity with optimal comfort for humans between 40 and 60%. Above this range, the temperature feels hotter, below this range it feels colder. Environmental temperature above or below thermoneutral values (e.g. between 17 and 28 degrees Celsius) both impact sleep in a similar way: resulting in an increase in sleep disturbances (arousals) and a decrease in sleep quality. Hotter ambient temperatures are more disruptive than cooler temperatures since the latter are easily modified by changing the sleep environment (additional blankets). Warmer temperatures within the thermoneutral range decrease sleep onset latency while the cooler range enhance deep sleep. Men in particular show an increase in sleep quality with temperatures at the lower limits of the thermoneutral zone.

**3.4.4 Noise [29,30,86–91].**   There are many sources of environmental nocturnal noise, but only specific categories of noise have been subject to studies. For many people in urban areas traffic, including road, rail and air, is a major source of nocturnal noise. The noise may differ with the type of traffic, but the effect on sleep is similar for all modalities as it leads to an increase in sleep disturbances and more wake time after sleep onset. Air traffic in addition impacts deep sleep negatively.

Wind turbine noise predominantly exists of low frequency noise: it can travel longer distances and penetrates further into building structures compared to high frequency noise. The impact of wind turbines on sleep quality is however ambiguous. A review on recent publications (2017–2020) on wind turbine noise, with a sound usually below 45dB, concludes that there is inconsistent and insufficient evidence for an effect on measurable markers of sleep such as sleep latency and sleep duration [90]. Wind turbine noise does, however, impact self-reported sleep quality such as perceived sleep disturbances.

Ambient noise consists of non-traffic and non-industry noise such as neighborhood noise, leisure noise, animal sounds or ringing bells. The outcomes on the relation between ambient noise and sleep show that meaningful sounds like leisure and neighbor noise, increases sleep disturbances whereas non-meaningful sounds like the humming of air-conditioning show ambiguous outcomes. Self-reported measurements, however, show the higher the level of low-frequency noise, the higher the effect on sleep. In general, animal sound showed no effect on sleep, whereas bell and recycling sounds showed a small detrimental effect.

**3.4.5 Light [82,92–95].** The evidence on the effect of light on sleep is conflicting. A review on dynamic light (i.e. artificial light that varies in intensity and/or spectral power) concluded that more light is impacting human performance but no final conclusions could be drawn regarding an effect on sleep [94]. One extensive review found a positive relation between light exposure and sleep in general but mostly ambiguous results when looking in more detail to the amount and timing of light and the effect on specific sleep parameters [95]. Bright light at night seems to result in lower sleep quality whereas there is a dose response relationship with the brightness and duration of the light exposure: the more exposure, the lesser quality of sleep. Bright light during the day seems to be reflected in improved self-reported sleep quality but the results on tracked sleep parameters are conflicting.

**3.4.6 Green space [96].** Increased exposure to green space (i.e. various forms of vegetation) during daytime lowers the risk on short sleep duration or insufficient sleep quality.

Table 5 provides an overview of all identified environmental determinants of sleep.

## 3.5 Personal and socio-economic determinants of sleep

Personal and socio-economic determinants consist of social and economic influences on sleep: in other words, the conditions in which people are born, grow, live work and age that reflect on their sleep. Though the health classification of determinants originally labeled this category as "socio-economic determinants", the elaboration of this category (e.g. social, cultural, and gendered roles, religious belief or spirituality, health cognition; and other societal contributors such as social attitudes, class/caste systems, community involvement and social and support systems) leaves room for including psychological-related determinants and makes it the most appropriate category within this classification. The personal and socio-economic determinants therefore not only include the social-economic situation of an individual but also include individual and personality related social concepts such as psychological dispositions. Sleep is a fundamental biological activity but can also be perceived as a complex social process in which the characteristics of the individual interact with the social environment.

**3.5.1 Attachment style [97].** Attachment style is measured along the dimensions of avoidance and anxiety. High anxiety has a negative impact on sleep quality and leads to differences in sleep architecture. This effect is stronger when high anxiety is combined with high avoidance as compared to low avoidance i.e. when people show socially avoidant behavior and are fearful of intimacy.

**3.5.2 Sexual orientation [98].** Sexual minorities include people with a lesbian, gay or bisexual identity. People who are part of a sexual minority, either in identity or actual behavior, experience in general shorter sleep duration and less sleep quality. Most sexual minorities experience greater stress through stigma and discrimination and through stress in family relations. Both stressors negatively impact sleep.

**3.5.3 Psychological dispositions [26,99–102].** Several psychological dispositions can affect sleep such as positive and negative affect, mood, humor and laughter, self-compassion and worry and rumination. Positive affect is associated with feelings of joy, happiness, energy or enthusiasm whereas negative affect is associated with feelings of sadness or anger. Both positive and negative affect are umbrella terms and may contain a broad range of different emotions and definitions: it may refer to a stable, enduring disposition or an aggregation of momentary ratings during the day. The impact of daily negative mood on the following night sleep is not significant, but there may be a cumulative effect over time: prolonged negative mood states may result in poorer sleep quality. High ratings of positive affect are associated with higher self-reported sleep quality, but not with tracked sleep parameters. Positive affect may act as moderator either accentuating or attenuating impact of risk factors on sleep

(protective effect). Very high levels of positive affect may have a negative impact on sleep. A review on laughter and humor showed no significant effect on sleep for healthy adults whereas for clinical populations there was a positive effect on sleep quality.

Self-compassion is a positive attitude towards oneself during difficult times to reduce suffering. Self-coldness can be defined as being aggressive or harsh towards oneself in moments of failure or suffering. There is a small positive association between self-compassion and sleep quality and a stronger negative association between self-coldness and sleep quality.

Self-compassion may play a protective role on sleep quality in reversing the negative effect of stressors and diminishing the impact of other factors such as perceived stress, rumination and anxiety.

Worry can be defined as negative repetitive thoughts about anticipated future events, whereas rumination describes negative repetitive thinking about events in the past. Both psychological dispositions have a negative association with sleep duration and sleep quality and a positive association with sleep latency (increase).

**3.5.4 Ethnicity [103–105].**    Several studies on the difference between African American and Caucasian Americans show that normal-sleeping African Americans do not sleep as well as Caucasian Americans as in general they experience more light and less deep sleep, longer sleep latency and less sleep quality. The effects are stronger for women and disappear with increasing age. When considering income, education, health status and employment, the ethnic differences diminish for low social economic status (SES) but remain significant for moderate and high SES.

There are limited studies on minorities and sleep, especially on intra-minority sleep. In general minorities are more likely to experience shorter sleep duration and less sleep quality whereas minorities (especially black people) are less likely to express sleep complaints.

**3.5.5 Work [27,106–112].**    The effects of some specific work characteristics on sleep have been studied such as job demand, job control and social support at work. Job demand can be defined as the demands and stressors associated with the job. A higher perceived job demand results in a decrease in sleep quality. Job control is the perceived ability to change your own work characteristics and the impact of job control on sleep is ambiguous.

Positive support at work from leaders impacts sleep positively by decreasing sleep disturbances and sleep latency. Positive support from co-workers has no significant effect on sleep. Negative support at work may decrease sleep quality and total sleep time and increases sleep disturbances.

Not surprisingly, being at work during normal sleeping hours, in many cases resulting from shiftwork, has an impact on sleep. The impact however differs with the timing of the shift. In general shiftwork reduces sleep duration but for evening shifts, starting between 2 and 5 pm, sleep duration is prolonged. The reviews on nightshifts are contradictory on sleep duration. Nightshifts result in longer sleep latency and increased WASO whereas sleep efficiency showed no difference.

A rapid rotating shift (i.e. a different shift in 4 days or less) results in more sleep disturbances. When working on call the outcomes on sleep are ambiguous on various sleep parameters, except for sleep duration: all reviews show a decrease on this parameter.

Workplace bullying refers to systematic exposure to harassment and non-physical mistreatment at work over a long period of time, usually in a gradually escalating situation. Workplace bullying is rather common, approximately 1 out of 10 people is a victim of workplace bullying in the Netherlands [113] and in the US this percentage is even 15% of the US workforce [109]. Workplace bullying is related to problematic sleep and shorter sleep duration. Important mediators for the effect on sleep are vulnerability to rumination and work-related worries, distress and the personal need for recovery. When there is physical violence involved at work, the

sleep problems are worse: less quality of sleep, longer sleep latency and an increased wake after sleep onset, potentially evolving into insomnia. Violence may be due to criminal intent but also includes customer-worker violence, worker-worker violence or the transfer of violence to work from the personal environment. In all studies the experience of violence was related to a previous period while the sleep problems continued over time.

**3.5.6 Psycho-social stress [114–116].** A psycho-social stressor (event) itself is not what causes stress but the perception of the stressor is [115]. The individual coping abilities determine to what extent the stressor impacts sleep. Daily life stress, characterized as chronic and non-severe stress, has a wide variety of potential stressors ranging from job stress to marital disruption. The effects of daily life stress on sleep are inconsistent, but all daily stressors lead to poorer sleep and an increase in awakenings. Stressful life events such as bereavement or divorce show that these are associated with poor sleep and usually negatively impact several sleep parameters such as sleep latency, the amount of REM-sleep, deep sleep and awakenings, depending on the life event and the individual. Experimental (acute) stressors show more consistency with increased sleep latency, more awakenings, decreased deep sleep, REM-sleep and sleep efficiency.

In general discrimination, a rather specific kind of social stress, has a negative effect on sleep. When looking into detail at the effect of discrimination on specific sleep parameters, the outcomes are however inconsistent.

**3.5.7 Social relations [117–124].** Social relations, including romantic, family and work relationships, and loneliness have a bidirectional relation with sleep. Positive social support has in general a favorable effect on sleep with more evidence for self-reported sleep outcomes than tracked outcomes. Negative social support has the opposite effect in diminishing sleep quality and increasing sleep disturbances. An effect on other sleep parameters was not available due to the limited number of studies.

When co-sleeping as a couple compared to solo sleep, sleep is affected on several parameters: shorter latency, more deep sleep, less REM-sleep. The higher the quality of the relationship, measured by attachment style and marital harmony, the better sleep quality and less sleep problems. A positive romantic relationship is associated with better sleep quality and a decrease in total sleep time whereas a negative relation (e.g. abusive or aggressive relation) leads to less sleep quality and more sleep disturbances. Note that it is suggested that providing partner support may cost for one's own sleep (duration) [122].

In a family setting the parent-child sleep is bidirectional as well and may evolve to a cyclical pattern. There is a medium effect size for the relation between parental and child sleep disturbances, leading to a decreased sleep quality for the caregiver, regardless of the child's age (when over two years old). The relation is stronger for maternal compared to paternal relations and includes an effect on sleep duration, sleep efficiency and awakenings. Given the bidirectional relation between parent and child sleep, the authors propose to consider sleep in a family context rather than in isolation.

Loneliness is defined as a mismatch between desired and actual social relations [125]. Different sleep parameters are mentioned in the reviews on loneliness: a strong association was found for loneliness and insomnia, a medium association for sleep quality and sleep disturbances and no association for sleep duration. The effect is stronger for men compared to women. In reverse, social participation leads to an increase in sleep quality, especially for women.

**3.5.8 Socio-economic status [126].** Socio-economic status is a marker of living conditions and habits that may influence sleep via stress related mechanisms. Barriers to accessing good living conditions such as quality of housing, social support and health insurance, influence the level of stress. The markers of socio-economic status are education, income, occupation and wealth. A lower socio-economic status is associated with shorter sleep duration and less sleep

quality. Difficult socio-economic position leads to an increase in wake after sleep onset. A higher socio-economic position leads to an increase in sleep duration, sleep efficiency and less sleep latency. The impact of the socio-economic status on sleep is mediated by individual health.

**3.5.9 Seasonal and cultural patterns [127].** During autumn and winter people tend to sleep longer whereas during spring and summer sleep duration is shorter. However, when comparing sleep during the seasons to the yearly average, no significant changes were found. In this review, Ramadan was considered a cultural pattern as well and again no significant changes in sleep duration were found.

Table 6 provides an overview of all identified personal and socio-economic determinants of sleep.

## 4. Discussion

The present umbrella review provides an overview of the current evidence on those determinants of natural adult sleep that have been previously covered by adequate systematic reviews. We found a wide variety of sleep determinants that were clustered around four main categories: biological, behavioral, environmental and personal/socio-economic determinants. Biological determinants are practically impossible to change but they can explain differences between individuals and life phases and provide as such valuable information on natural sleep developments. Research on environmental determinants is mainly limited to specific categories such as light and noise. Personal and socio-economical determinants impact sleep directly by positive social relations and positive psychological dispositions. Moreover, they indirectly impact sleep by increased stress levels due to social and personal circumstances. Behavioral determinants provide the most promising approach to improve sleep though it is complex, as behavior in general has many parameters and the effect changes with the specifics of these parameters.

Studies on determinants of natural sleep are a complex, but highly fragmented field of research, as no review combined more than one category of determinants. Almost all reviews presented a different level of aggregation, different target groups and outcomes on different parameters of sleep. In addition, almost all determinants appeared to have different dimensions leading to different outcomes on sleep, e.g. physical exercise has the dimensions type, duration, intensity and regularity of training. To add complexity, several determinants interfere with another in relation to sleep, e.g. mood moderates the relation between physical activity and sleep, whereas mood regulation itself can be classified as a determinant of sleep [57,61,62]. To put in summary, sleep is a highly complex phenomenon and when studying sleep determinants, we should value its complexity instead of limiting interpretations to unambiguous definitions of sleep parameters or determinants.

Both definitions of the sleep parameters and the measurement methods show limited consistency; the measurement methods vary from tracked to self-reported, from questionnaires to indepth interviews to actigraphy. Though we made a distinction between sleep quality and sleep duration, the difference between subjective and objective types of sleep measurement is not as straight-forward to be able to differentiate between both and therefore we used the terminology self-reported and tracked instead. In general, only one single night was studied, while the necessity to study several consecutive nights to diminish the intra-individual variability, i.e. small daily variances in sleep duration or sleep quality, is well-acknowledged [40,128]. In fact, the intra-individual variability could be classified as both an extra dimension on other sleep parameters, as well as a separate determinant, i.e. (ir)regularity of sleep, impacting sleep quality [40].

The night is a reflection of the day, meaning that daytime behavior and daytime experiences impact sleep by so called somnoprints. A somnoprint is a characteristic sleep pattern, related to events occurring during waking [129]. Sleep itself is not an example of conscious behavior.

Though sleep is an active process and interlinked with daytime activities, it is rather the outcome of a range of autonomic processes and deliberate actions prior to sleep itself. The way in which determinants of sleep impact these autonomic processes and conscious behavior is yet to be unraveled, but it is clear that they do impact sleep itself. These deliberate actions preceding sleep are known as sleep hygiene: sleep promoting behaviors, which are widely known and accepted as rules of thumb when dealing with sleep deprivation. The determinants of sleep that are identified in this umbrella review provide a complex and broad picture, whereas sleep hygiene rules are rather the opposite: clear, easy to understand and limited to the most promising options to improve one's sleep. Despite the fact that promoting sleep hygiene rules are common practice, several articles challenge these practices: better sleep hygiene awareness does not guarantee better sleep quality probably because sleep hygiene is not found useful or used wrongly [130]. Another review concluded that sleep education increased sleep knowledge but studies towards its effect on actual sleep parameters showed mixed results [131]. In addition, Irish a.o. [132] state that the sleep recommendations, though theoretically sound and plausible, are somewhat vague and inconsistent and have limited use for natural sleep behavioral patterns. They plead for more specific guidelines, a broad spectrum of sleep hygiene behavior and a more individualistic implementation of sleep hygiene rules to support individuals in sleep improvement.

Most reviews mention several confounders, moderators and/or mediators but fail to make a clear distinction between these constructs. From the perspective of the health practitioner, the distinction between confounders, moderators and mediators could be useful to differentiate between highly promising (i.e. easily accessible and changeable moderators) versus less promising (i.e. confounders) starting points for sleep interventions and actions. For example, regulating your core body temperature by a lukewarm shower after an evening workout is fairly easy compared to changing your overall health status. Regarding weight (or BMI) it should be noted that overweight has in many cases a direct relation with obstructive sleep apneas, thereby severely impacting sleep quality. Health practitioners supporting overweight patients should address weight control as a first line of action.

From the perspective of the promotion of healthy sleeping, it is important to know which sleep determinants are modifiable and which are not. Genetics or an airport in the neighborhood are practically non-modifiable as opposed to eating or drinking behavior. Furthermore, knowing which changes in sleep parameters can be expected when applying sleep hygiene rules provides valuable information. For example, a diminished sleep onset latency is quite different from experiencing less sleep disturbances. In addition, various determinant-specific moderators that were identified in the present umbrella review, may potentially diminish or increase the impact of an intervention. For example, the violence level of a video game or the perceived intensity of the physical activity that is being promoted.

Including important moderators in the sleep hygiene rules, relevant for the individual in his or her context, make them more personalized and improve the potential effectiveness related to sleep.

In conclusion, extending the sleep hygiene rules with the identified sleep determinants, the related sleep parameters and important moderators from this umbrella review could help counsellors and their clients to become aware of their personal set of relevant determinants and identify promising starting points for sleep improvements.

## 4.1 Limitations

Sleep is a hot topic in research and potentially new determinants might emerge rapidly. However, this comprehensive umbrella review, covering a wide variety of determinants, provides

an overarching starting point that leaves room for add-ons. Selecting the sleep determinants from published reviews results in well substantiated factors that influence sleep. On the other hand, this umbrella review does not reflect a balanced overview of all determinants since there will be a lack of reviews and meta-analyses on determinants of sleep that have been less subject to research. The exclusion of reviews on children and adolescents, led to less attention for research on some determinants, such as social media use, which was mainly targeted at children and adolescents [133]. Similarly, research on age mostly included and focused on children and adolescents [134]. Potential additional determinants will provide further detailed information within a category of determinants. An important area of concern and a limitation is that determinants are interrelated, which means that the relationship between determinants and sleep outcomes is not strictly linear, but most likely more complex. This interrelatedness is one of the reasons we excluded studies on sleep interventions as it is difficult to single out the impact of one individual factor as a determinant. A major limitation is the fact that only part of the reviews was of sufficient quality. Additional research on sleep and sleep determinants will provide more insights in addition to those provided in the present review.

The level of aggregation of the included reviews differs significantly. In some cases, one review included a main category of determinants, e.g. environmental determinants of sleep, as opposed to another review that only included one specific determinant of sleep, e.g. artificial light at night and sleep. We included all levels of aggregation to combine as much information as possible. Again, the wide scope of the umbrella review and the outline in four main categories support further detailing on underlying determinants when identified.

Tracked and self-reported measurements are hard to compare as sleep perception differs from actual sleep quality [135] and self-reported sleep quality may be correlated with non-sleep phenomena such as health status, pain, mood or anxiety [21]. For practical reasons we combined self-reported and tracked data supported by the idea that from the perspective of the health professional both outcomes are important. Tracked data provide a clear picture on the (gravity of the) situation and determinants whereas the self-reported outcomes provide insights into the importance and experience as perceived by the individual. Above all, both points of view provide valuable leads to enter a conversation on sleep improvements.

## 4.2 Strengths

Sleep research is still fragmented. This umbrella review is the first to provide a comprehensive overview of all determinants on natural adult sleep. This umbrella review confirms existing sleep hygiene rules and adds scientific background and elaboration for practical use. In the Netherlands, for example, Combined Lifestyle Interventions (CLIs) focus on diet, physical activity and behavior change. As stated by the National Healthcare Institute (Dutch: Zorginstituut Nederland) "a CLI aims at behavior change to reach and sustain a healthy lifestyle by providing coaching and advice on diet, exercise and behavior change" [136]. According to the Partnership Overweight in the Netherlands (PON) treatment of the CLI consists of diminishing energy intake, increasing physical activity and optionally adding psychological interventions to support behavior change [137]. The guidelines of the Dutch College of General Practitioners (Dutch: Nederlands Huisartsen Genootschap) for the treatment of obesity also mention a.o. diet and physical activity, with a specific focus on alcohol [138]. None of these Dutch institutions mention the topic of sleep as major focus and consequently sleep is receiving less attention in the treatment of overweight/obesity. This is in strong contrast with the current empirical base that shows that sufficient sleep of good quality is considered a prerequisite for successful and sustainable lifestyle interventions and improving sleep (deprivation) is considered a necessity for weight control and/or weight loss [139].

The limited attention for sleep is reflected in training courses. Consequently, health practitioners have on average limited knowledge on sleep. While the examples above focus on the Dutch context, the same might hold true for practitioners in other countries. The umbrella review provides useful insights for practitioners when supporting people to improve their sleep, beyond simply applying general sleep hygiene rules.

### 4.3 Recommendations for future research

This umbrella review provides an overview of the current stand on research on sleep determinants, useful in day-to-day practice of health professionals addressing sleep issues with their patients. To enhance the practical use of the sleep determinants in daily practice, we recommend the presentation of the determinants in a well-structured and summarizing model, providing a subject for further research.

Since sleep research is a trending topic this umbrella review reflects the current availability of reviews on the determinants of sleep in natural circumstances but further research is to be expected. Given the broadness of the topic of sleep, we mainly address directions for further research instead of specific recommendations.

Our suggestions for future research are:

- More systematic reviews or meta-analyses on determinants that receive less attention, for example bedding and mattresses or cognitive activity, would greatly enhance the quality and usability of the overview.

- More effort into clear definitions of relevant sleep parameters.

- Several consecutive days should be considered when looking at sleep in research and practice.

- More research elaborating on the relation between the different sleep determinants and the effect on accompanying sleep parameters.

- More effort should be put into designing high quality intervention studies on sleep.

## 5. Conclusion

A comprehensive overview on relevant sleep determinants provides a practical and evidence-based starting point to identify relevant interventions to secure or improve individual sleep quality. Extending generic sleep hygiene rules with an overview of all types of potential determinants enhances the awareness on the complexity and importance of sleep and can be used to improve the effect of sleep interventions in health promotion.

## Supporting information

**S1 Checklist. PRISMA 2020 checklist.**
(DOCX)

**S1 File. Search strings.** This file contains the search string used for the selection of articles.
(PDF)

**S2 File. Overview of the data extraction.** This file contains the key elements of all selected articles in the umbrella review.
(XLSX)

**S3 File. Quality indication of selected articles.** This file contains an overview of the quality indication of all selected articles.
(XLSX)

## Acknowledgments

The authors like to thank Gregor Franssen for his advisory role in defining the search strategy for the umbrella review.

## Author Contributions

**Conceptualization:** Nicole Philippens, Ester Janssen, Stef Kremers, Rik Crutzen.

**Methodology:** Nicole Philippens, Ester Janssen, Stef Kremers, Rik Crutzen.

**Validation:** Nicole Philippens, Ester Janssen.

**Visualization:** Nicole Philippens.

**Writing – original draft:** Nicole Philippens.

**Writing – review & editing:** Nicole Philippens, Ester Janssen, Stef Kremers, Rik Crutzen.

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
