## [Decision Letter · Decision Letter 0]

3 Jun 2022

PONE-D-22-10437Determinants of natural adult sleep: an umbrella reviewPLOS ONE

Dear Dr. Philippens,

Thank you for submitting your manuscript to PLOS ONE. After careful consideration, we feel that it has merit but does not fully meet PLOS ONE’s publication criteria as it currently stands. Therefore, we invite you to submit a revised version of the manuscript that addresses the points raised during the review process. In particular, your manuscript was evaluated by four expert reviewers, whom I would like to thank and commend for the wealth of suggestions they offered to improve your manuscript and for their constructive criticism. While the reviewers found some merit in your work, they all raised important critical issues that prevent publication of your manuscript in its present form. These issues broadly concern PLOS publication criteria 3 (analyses  performed to a high technical standard and described in sufficient detail) and 4 (conclusions presented in an appropriate fashion and supported by the data). I understand that a sweeping major revision of your manuscript would be needed to address the reviewers' comments satisfactorily. Nevertheless, given the ambition of your work, I encourage you to perform such extensive revision taking advantage of the in-depth review, comments, and suggestions provided by the reviewers. Please also consider narrowing the focus of your review. 

We look forward to receiving your revised manuscript.

Kind regards,

Alessandro Silvani, M.D., Ph.D.

Academic Editor

PLOS ONE

Journal Requirements:

Reviewers' comments:

Reviewer's Responses to Questions

**Comments to the Author**

1. Is the manuscript technically sound, and do the data support the conclusions?

Reviewer #1: Partly

Reviewer #2: Partly

Reviewer #3: Yes

Reviewer #4: No

2. Has the statistical analysis been performed appropriately and rigorously? 

Reviewer #1: Yes

Reviewer #2: N/A

Reviewer #3: N/A

Reviewer #4: N/A

3. Have the authors made all data underlying the findings in their manuscript fully available?

Reviewer #1: Yes

Reviewer #2: Yes

Reviewer #3: Yes

Reviewer #4: No

4. Is the manuscript presented in an intelligible fashion and written in standard English?

Reviewer #1: Yes

Reviewer #2: Yes

Reviewer #3: Yes

Reviewer #4: No

5. Review Comments to the Author

Reviewer #1: In the paper “Determinants of natural adult sleep: an umbrella review” the authors aimed to provide an overview of the current evidence on determinants of natural adult sleep.

The purpose of the review is certainly of relevance and in a complex and, sometimes, fragmented landscape such as sleep medicine these efforts are to be maintained. However, several methodological issues are to be outlined and considered because the risk of overgeneralization is often around the corner.

The main issue in this case is in the selection of sleep measures considered upon which the determinants are evaluated. The authors divide them in two groups: sleep duration and sleep quality. However, while they outpoint that sleep quality may be a subjective matter and consider the individual perspective, they fail to discriminate the two faces of the same coin. First, objective “sleep quality” (OSQ) taken from recorded parameters is often inconsistent with subjective “sleep quality” (SSQ) evaluated through questionnaires and interviews. Therefore, the authors should clearly state what type of evidence it is used, a mix of the two would result in inappropriate conclusions. Indeed, some of the variables (such as REM latency) can only be grasped by OSQ evaluation, some by both (though, again with intrapersonal inconsistences) such as WASO and some only by SSQ (sleep quality sensu stricto). Moreover, the division between sleep duration and quality can be inconsistent too, as many of the variables of sleep quality (eg, WASO or NAASO) directly impact sleep duration. Eventually, the evidence should be divided in evidence from objective evaluation of sleep (polysomnography, actigraphy, wearables including EEG or actigraphy recording) and evidence from subjective evaluation.

Moreover, some other relevant points should be evaluated:

• In the Introduction section the authors refer to “artificially induced sleep as in study labs”. Please consider that no drug exists as of today that can induce actual sleep. Drugs that can lead to sleep or help sleep initiation are present, but sleep should be considered as a natural event and cannot be artificially induced (unlike other states of reduced or absent consciousness). This may sound intricate but should lead also lead to another consequence. Actual sleep recorded in a sleep lab (through polysomnography) is indeed natural sleep and should be taken into consideration. The impact of one night of sleep in a sleep lab instead of at home sleep may be present, but the objective features of healthy sleep are maintained and in lab sleep evaluations are efficiently used to evaluate how different determinants act over sleep. In conclusion, sleep lab evaluations although considered as a confounder should also be taken into consideration.

• On a practical point of view, the authors should better define how they excluded reviews referring to different registration methods of sleep, as, indeed, many of them are required to evaluate the objective parameters taken in consideration in the review. Another point should be the redefinition or the better definition of what is a determinant and what is a “sleep intervention” which was excluded by the authors. Why did for example the authors consider meditative activity (such as Pilates, tai chi) as determinants and meditation relaxation as an intervention? Many of the previous share similar features and indeed, may be performed by the individuals for the very same reasons.

• Please better define why some determinants were also treated as confounders at the beginning of the Results section, such as age, sex, BMI etc… These are indeed determinants of sleep and interweave with the others as much as for example work schedule and ethnicity do. It is true that being basic demographic features are the ones that are better evaluated among the other determinants, but the final level of impact on sleep is no different from the others, these a priori differentiation should not be made.

• The authors decided to include pain into the evaluation as a determinant of sleep in healthy people. However, pain is already a “unhealthy condition”, and cannot be discriminated or disentangled from the other diseases or illnesses. By trying to evaluate pain per se without the contribution of other morbid conditions, the authors underlined how pain does not affect sleep. This, in my opinion, leads to an incorrect assumption. The authors should either exclude it from the evaluation or decline it into the different conditions (eg, osteoarticular, neuropathic pain, etc…). In this case overgeneralization could have been detrimental.

• In the Discussion section the authors state that definition of sleep parameters may be inconsistent and that better efforts should be made into definition of sleep parameters. This point should be better tackled. Objective sleep parameters are defined, and no inconsistencies are present. The perceived inconsistencies may result in subjective evaluation, that may not find a solution as they disperse in the complex interindividual and intraindividual variabilities of perceived events. Sleep is a complex and relevant part of our lives and the inability to generalize its subjective aspects should not be taken as our lack of methodologies, but rather as a inevitable matter to deal with. To an extent we could say that generalizing sleep would be on the same level as generalizing individual determinants of daytime activities. When evaluating such complex issues as sleep determinants, and trying to analyze each determinant per se, the drive towards generalization should give way to the knowledge that complexity and its underneath interactions are just to take into account.

• It is incorrect to define sleep as an “outcome of a range of autonomic processes and deliberate actions prior to sleep itself”. Sleep is a active process, and not the passive bystander and consequence of daytime activities and “determinants”. Sleep can directly affect daytime activities and its regulation go beyond the autonomic processes undergoing its phases. The relation sleep-daytime life is to be considered as biunivocal and at the same level, not with one as a byproduct of the other.

• In the case of “disrupted sleep”, sleep hygiene is the first line approach to be taken into consideration. Sleep hygiene rules are simple, but effective, because they are the very bases, the foundation of good sleep. The fact that sleep is complex with many determinants is not a contradiction, while the authors seem to point out so. To make a parallelism with diabetes, it is true that diabetes may have various complex determinants, but if you do not follow the very simple rule of moderating sugar intake your diabetes will worsen. That is the same. Moreover, the fact sleep hygiene is not always effective should not be a reason for discouraging. For example, for primary insomnia, if the patient does not follow these rules and does not really accept that his/her will to change the situation is their primary determinant for the condition, a solution could not be found. That would be equal to saying that we should not always suggest some drugs because of some patients scarce compliance.

As a final remark and linked to the matter of complexity-simplification, the very strict criteria which finally cut-out the majority of evidence, actually oversimplified the matter. The pruning of evidence was to an extent, excessive, and the final corpora resulted impoverished rather than better specified. This was probably exemplified by the fact that some determinants actually had only one review they were taken from, which may be a source of bias for the umbrella review.

Reviewer #2: The manuscript describes an umbrella review aimed at providing an overview of the current evidence on determinants of natural adult sleep. The authors conducted a literature search on six electronic databases (PubMed, WoS, Embase, CINAHL, PsycInfo and Cochrane), used a shared coding system for assessing quality of the selected articles (AMSTAR2 tool) and registered the review on PROSPERO. Ninety 3 reviews and meta-analyses were identified. Results evidenced that each determinant was found to affect different sleep parameters and the relationship with sleep is influenced by both generic and specific moderators.

The topic is worth to be studies, however several pitfalls should be acknowledged.

Introduction

The authors start their introduction traying a definition of sleep. This is a tall order issue. In 2007 Chokroverty discussed in details the history of sleep research, evidencing that starting from the 40ths and 50ths of the last century to his present, research on the neuro-psychophysiology of sleep has advanced greatly but 2 basic questions were still open: What is sleep? and Why do we sleep? These questions are still almost open and each researcher that addresses sleep issues, usually skips its definition, preferring to adopt a descriptive approach and thus addressing its characteristics or structure or “determinants”.

Instead the authors chose to start their introduction giving a questionable definition: sleep is “a reversible and repetitive condition of diminished consciousness” using as reference Laar et al 2021 (Laar Mvd, Hadden B. Slapen als een oermens : wat de evolutie ons leert over een goede slaap. Eerste druk. ed. [Eindhoven]: Merijn van de Laar; 2021) that is in a language not accessible to all international readers. I suggest to give a descriptive definition (see for instance Hirshkowitz, 2004 “Sleep can be defined many ways; however, the basic core concepts remain the same. First and foremost is that sleep is a brain process. The body rests but the brain sleeps. This is not to say the body does not require sleep; there are essential body processes that occur only when the brain is asleep. Nonetheless, the brain is what does the sleeping. The second core

concept is that sleep is not a unitary phenomenon.”).

Reference 11 is very old. Many new papers could be cited for acknowledging updated effects of sleep deprivation on executive functions, mood, autonomic function, immune system, job performance and risk of traffic or industrial accidents.

Moreover there is a consensus report about how much sleep do we need that could be cited (Watson NF, Badr MS, Belenky G, Bliwise DL, Buxton OM, Buysse D, Dinges DF, Gangwisch J, Grandner MA, Kushida C, Malhotra RK, Martin JL, Patel SR, Quan SF, Tasali E. Recommended amount of sleep for a healthy adult: a joint consensus statement of the American Academy of Sleep Medicine and Sleep Research Society. SLEEP 2015;38(6):843–844).

There is a logical leap, at the end of the introduction, where the authors simply state that “Sleep is typically conceptualized broadly, containing different aspects when operationalized, so called sleep parameters (e.g., sleep duration, sleep quality, sleep disturbances or even some aspects of dreaming). As a result, sleep has a broad range of determinants”. This leap introduces the determinants of sleep that are the focus of the review while all the introduction focuses on the importance of sleep. I suggest to shorten the previous part of the introduction and to give more space to discussing why sleep may be operationalized through its different parameters and what does each parameter means. These aspects are addressed in the method section but actually they have a theoretical background that may be introduced in the introduction section.

Results section

“Socio-economic determinants consist of social and economic influences on sleep: in other words, the conditions in which people are born, grow, live work and age that reflect on their sleep. The socio-economic determinants not only include the social-economic situation of an individual but also include individual and personality related social concepts such as psychological dispositions”.

Considering personality, attachment style, sexual orientation and psychological dispositions in general as socio-economical determinants is questionable and the authors do not argument on it nor report on which bases they made that decision. It is my opinion that psychological characteristics should be distinguished from socio-economic characteristics. Moreover, the reference the authors cite for supporting their categorization (i.e. Health NSWDo. Public Health Classifications Project–Determinants of Health. Phase 2 Two: Final Report. 2010) does not include psychological characteristics/dispositions within the socio-economic determinants of health.

Reviewer #3: I must say I have quite appreciated this umbrella review aimed to summarize evidence from meta analyses and systematic reviews on the impact of the main determinants of natural sleep

In fact, I would like credit quite many aspects: good organization, clear explanation of methods, rigorous approach to data search, well-balanced theoretical comments in the discussion including useful suggestions and rules of thumbs for applicative/clinical purposes.

Still, there are a few major issues, plus some minor points that could be addressed and improved throughout the paper, that I would definitely encourage the authors to reconsider in light of the comments that I am adding below:

Major issues:

1.First of all, I suggest that some cautionary comments are made on the intrinsic limit of the “umbrella review” approach, that unavoidably suffer from a literature bias. What I mean is that some determinants will be overestimated because more reviewed in the literature whereas others, often even more important, will be neglected due to the partial/total lack of reviews (not necessarily corresponding to a lack of experimental research). More specifically, in this article there are some extremely relevant determinants of sleep which have received very little space (e.g., cognitive activity, gender) as compared to rather minor determinants (meditational physical activity, music, intermittent fasting), and some other are completely absent (e.g., electromagnetic fields are among the most debated environmental determinants, and bedding habits, such as posture, mattresses eccetera, are also not mentioned).

2.At this regard, I wonder whether the paragraph on age does really cover all the reviews published on the topic. By heart, I remember at least two that are pioneering and extremely important: Bliwise DL, Sleep 1993, 16(1):40-81; Ohayon MM et al., Sleep, 2004;27(7):1255–1273. What is the reason for their exclusion?

3.I am also kind of surprised by the assessment of the analyzed reviews’ quality (Page 10 and table 2). Although the method used is appropriate and well-described, I am negatively impressed by the very low proportion of determinants supported by “sufficient quality” literature (5 out of 29, namely 0 out of 3 for biological, 2 out of 11 for behavioral, 1 out of 6 for environmental, 2 out of 9 for socio-cultural). In my opinion, the rating could be too conservative, given that RoB is very seldom addressed even in allegedly excellent reviews that provide a precious contribution (I know some of them myself among the cited papers). Therefore, one option would be to change the “sufficient, mediocre, questionable” scale in “good, sufficient, questionable”. Otherwise, it would be necessary to report the general low quality of the available literature, in the Discussion, as a major limitation of the study.

4.I think the authors should try to be somehow clearer on the definition of sleep quality, which is an extremely tricky matter. First, objective sleep measures traditionally considered as indices of sleep quality (such as, for example, sleep depth and sleep continuity), do not utterly correspond to the determinants of subjective sleep perception, including aspects such as feeling refreshed at awakening. So the idea of collapsing in a single measure both objective and subjective sleep quality is questionable. Secondly, there is a close relationship between different determinants of sleep quality. Clearly, I realize that some sort of generalization is needed to compare different data sources and to get an overall view, but the theoretical problem should be addressed in more details and represents a limitation that should be definitely mentioned.

5.Also, talking of interconnections between variables, I have not fully understood how the data have been treated concerning interdependent determinants. For example, how can we disentangle what is said about the effects of chronotypologies from the well-known evidence that they show consistent age-related changes? What about the possibility that socio-economic status is reflected from varying living conditions (i.e., from modifications of the environmental determinants)? I suppose some explanation is provided at page 12, lines 252-254, but it frankly seems very unclear to me.

6.Physical activity and sedentary behavior are two sides of the same medal. Also, video gaming is not a specific type of sedentary behavior. Therefore, I would definitely combine them in the same paragraph, keeping social media and video games separated.

7.Finally, there is some sort of confusion on the notion of “natural sleep”. In the Inclusion and Exclusion criteria section, the authors state (page 7, lines 148-151) that “the focus on healthy natural sleep led to the exclusion of reviews on a specific disease population, illness-oriented reviews on sleep (e.g. sleep and diabetes, asthma, eating disorders, HIV, or AIDS), sleep diseases (e.g. bruxism, narcolepsy, sleep apnea) or on excess behavior and addictions (e.g. problematic smart phone use, alcoholism)”. I can agree on this choice, but this is somehow contradictory with the insertion of “pain” as a biological determinant and, above all (page 11, lines 239-242), of sleep medication as a mediator (since sleep medications exclude the notion of natural sleep by definition).

Minor issues:

the notion of “sleep determinants” could not be clear to all readers. Please, give a short definition already in the abstract (page 1, line 15)

The “results” section in the abstract is too generic. Please, identify and mention either the most impacting determinants or the most important variables that are affected.

Page 2, line 30: a full stop point is missing between “review” and “Extending”

Page 2, line 39: Either remove the full stop point between [2] and “Whereas" or replace “whereas” with “Instead,”

Page 2, line 47: Replace “less “ with “decreased” or “reduced” and replace “is” with “are”

Page 2, line 47: Is there really a “causal” link between sleep disturbances and diabete? In the abstract of the quoted paper (Reutrakul and Van Cauter, 2014) it is stated that “Several large prospective studies suggest that these sleep disturbances ARE ASSOCIATED with an increased risk of incident diabetes”. Please, verify.

Page 3, line 64: “ (…) sleep duration of 90% of Dutch adults is in accordance with the recommendations of the American Association of Sleep Medicine (AASM)”. Please, specify what is this recommendation about sleep duration.

Page 3, lines 67-69: “A poll (…) enough time to sleep [15].” Confused sentence, please rephrase.

Page 3, line 72: “The results (…) United Kingdom [15]” Are these all surveys on habitual sleep?

Page 4, line 75: Please, erase “one’s”

Page 4, line 78: “sleep disturbances” and “aspects of dreaming” are not sleep parameters.

Page 4, line 82: please, replace “with” with “according to”

Page 6, Table 1: right column, second row (SOL): erase “with which”

Page 10, lines 218-220: “We could not (…) determinants”. Unclear sentence, please rephrase.

Page 11, line 225: please, replace “around before mentioned” with “around the before mentioned”

Page 11, line 226: what does “the nature of the relation with sleep” mean?

Page 11, line 233: please, replace the full stop after weight with a comma.

Page 12, line 246: please, replace “mental” with “psychological”

Page 12, lines 249-250: "Biological determinants(…) for an individual”. Incorrect, see for instance age and chronotypologies

Page 12, “Age and sex” section: the age range of the umbrella review should be specified, either here or in the Method. What is the younger and the older age included?

Page 12, lines 263-264: “However (…) men.” Good point! This distinction between subjective and objective sleep quality should be kept in mind also elsewhere (see major issues, comment n. 4)

Page 12, lines 265-266: “Elderly (…) deep sleep.” In the comparison with men? Or with other ages? Or relative to other stages? Please, specify.

Page 12, line 266: Please, replace “on” with “of”

Page 13: please, add “about” between “when” and “individual” and replace “or when they are most alert” with “and when they prefer to stay awake”

Page 13, line 277: is it “subjective” sleep quality?

Page 15, line 307: what kind of “sleep disruptions”?

Page 16, lines 334-335: “For older (…) impact on sleep”. Unclear please rephrase

Page 20, lines 424-425: “(Physical) (…) change in sleep”. I do not understand the meaning and implication of this sentence

Page 33, line 666: “on determinants”. Please, specify “on those determinants that have been previously covered by adequate systematic reviews”

Page 33, line 674: “longevity” ???

Page 34, lines 690-691: Unclear sentence, please rephrase.

Page 34, lines 704-708: Excellent point!

Reviewer #4: I read with interest the manuscript entitled “Determinants of natural adult sleep: an umbrella review.” In this study Philippens and co-authors aimed at providing an overview of the determinants of natural sleep by conducting an “umbrella review” i.e., a meta-review based on previously published reviews and meta-analyses. Their final goal was to provides a practical, scientifically-based, background to develop novel interventions aimed at improving sleep quality.

The authors identified and analyzed 93 reviews and meta-analyses. Results were categorized in four main categories: biological, behavioral, environmental, and socio-economical determinants. Quality of the selected articles was assessed using a method based on the AMSTAR2 tool.

Overall, the authors concluded that each determinant affect different sleep parameters although with a high degree of overlap.

The study aim, while relevant, is extremely ambitious, far too ambitious to be addressed in a single manuscript. The research question is too broad, and the authors failed to provide a comprehensive nor "practical" picture for any of the specific determinant of sleep under investigations.

Here some suggestion to improve the manuscript

1. The topics covered are far too broad. This issue has an important influence on the degree of analytical depth with which the specific determinants are analyzed. Indeed, the results described are very academic and does not add anything new to the knowledge already possessed by clinicians and health professionals working in the field of sleep. I strongly suggest narrowing down the review to a maximum two determinant: biological and behavioral determinants of sleep are the most relevant topic to cover with the aim of developing novel interventions on the other hand environmental, and socio-economical determinants are far less studies and could be more interesting to review.

2. The review was not performed according to PRISMA guidelines.

While this specific type of review may not be covered by the PRISMA guidelines, it would have helped the authors define, and subsequently analyze, a more precise research question.

3. The manuscript its present version, runs the risk of being a sort of "shopping list" of available results. The authors are strongly encouraged discuss results more proactively by adding a few lines of reasoning that help a general recap. The Discussion section could be enriched by discussing more explicitly and extensively some theoretical aspects.

4. The manuscript would really benefit from a “Research Agenda/Future Prospective” section.

6. PLOS authors have the option to publish the peer review history of their article (what does this mean?). If published, this will include your full peer review and any attached files.

Reviewer #1: **Yes: **Luca Baldelli

Reviewer #2: No

Reviewer #3: No

Reviewer #4: **Yes: **Marco Filardi

---

## [Author Response · Author response to Decision Letter 0]

4 Jul 2022

REVIEWER 1

In the paper “Determinants of natural adult sleep: an umbrella review” the authors aimed to provide an overview of the current evidence on determinants of natural adult sleep.

The purpose of the review is certainly of relevance and in a complex and, sometimes, fragmented landscape such as sleep medicine these efforts are to be maintained. However, several methodological issues are to be outlined and considered because the risk of overgeneralization is often around the corner.

Response: Thank you for confirming the relevance of this umbrella review and for providing us with such valuable and constructive feedback.

Comment: (1.1) The main issue in this case is in the selection of sleep measures considered upon which the determinants are evaluated. The authors divide them in two groups: sleep duration and sleep quality. However, while they outpoint that sleep quality may be a subjective matter and consider the individual perspective, they fail to discriminate the two faces of the same coin. First, objective “sleep quality” (OSQ) taken from recorded parameters is often inconsistent with subjective “sleep quality” (SSQ) evaluated through questionnaires and interviews. Therefore, the authors should clearly state what type of evidence it is used, a mix of the two would result in inappropriate conclusions. Indeed, some of the variables (such as REM latency) can only be grasped by OSQ evaluation, some by both (though, again with intrapersonal inconsistences) such as WASO and some only by SSQ (sleep quality sensu stricto). Moreover, the division between sleep duration and quality can be inconsistent too, as many of the variables of sleep quality (eg, WASO or NAASO) directly impact sleep duration. Eventually, the evidence should be divided in evidence from objective evaluation of sleep (polysomnography, actigraphy, wearables including EEG or actigraphy recording) and evidence from subjective evaluation.

Response: We agree that the distinction between different tracking methods of sleep measures would provide valuable insight. Differentiating between objective and subjective is however difficult and not as straight-forward as needed to make the distinction. That is the main reason why we used the terms “self-reported” and “tracked” instead. In addition, most reviews contain a combination of both measuring methods and tracking and displaying the sleep measuring methods for each underlying study of the included reviews does not fit the purpose and working method of the umbrella review. 

Given the valuable comment, we have incorporated an additional comment in the discussion section of the revised version of our manuscript on this topic: ‘Though we made a distinction between sleep quality and sleep duration, the difference between subjective and objective types of sleep measurement is not as straight-forward to be able to differentiate between both and therefore we used the terminology self-reported and tracked instead.’ (page 34, line 802).

Comment: (1.2) Moreover, some other relevant points should be evaluated:

In the Introduction section the authors refer to “artificially induced sleep as in study labs”. Please consider that no drug exists as of today that can induce actual sleep. Drugs that can lead to sleep or help sleep initiation are present, but sleep should be considered as a natural event and cannot be artificially induced (unlike other states of reduced or absent consciousness). This may sound intricate but should lead also lead to another consequence. Actual sleep recorded in a sleep lab (through polysomnography) is indeed natural sleep and should be taken into consideration. The impact of one night of sleep in a sleep lab instead of at home sleep may be present, but the objective features of healthy sleep are maintained and in lab sleep evaluations are efficiently used to evaluate how different determinants act over sleep. In conclusion, sleep lab evaluations although considered as a confounder should also be taken into consideration.

Response: We certainly agree with the reviewer that sleep itself is a natural event, even when sleeping in a lab. However, we explicitly chose to evaluate sleep in natural circumstances to support the practical use of the knowledge on sleep determinants in day-to-day practice of health professionals in a primary care setting. Therefore, we decided to clarify this in the definition of natural sleep used throughout the manuscript: ‘Throughout this manuscript we refer to natural sleep as sleep in natural circumstances (e.g. sleep encountered in everyday life)’ (page 2, line 45).

Comment: (1.3) On a practical point of view, the authors should better define how they excluded reviews referring to different registration methods of sleep, as, indeed, many of them are required to evaluate the objective parameters taken in consideration in the review.

Response: We did not exclude reviews based on the registration method of sleep. To clarify this, we incorporated the following explication in the method section of our revised manuscript: ‘All tracked and self-reported sleep registration was included’ (page 6, line 164).

Comment: (1.4) Another point should be the redefinition or the better definition of what is a determinant and what is a “sleep intervention” which was excluded by the authors. Why did for example the authors consider meditative activity (such as Pilates, tai chi) as determinants and meditation relaxation as an intervention? Many of the previous share similar features and indeed, may be performed by the individuals for the very same reasons.

Response: We agree with the reviewer that our definition of what is a determinant and what is an intervention left room for discussion. Therefore, we incorporated the following definition in the method section of the revised manuscript: ‘A (behavioral) determinant describes daily routines not necessarily linked to (improve) sleep (which a person normally does) whereas an intervention is specifically targeted at improving sleep (which a person is instructed to do within the context of an intervention). We excluded interventions in line with the objective to study sleep in natural circumstances’ (page 8, line 213). 

It goes without saying, that after an intervention, some of those practices during the intervention might become daily routine, but the focus here was not on the effects of interventions as such.

Comment: (1.5) Please better define why some determinants were also treated as confounders at the beginning of the Results section, such as age, sex, BMI etc… These are indeed determinants of sleep and interweave with the others as much as for example work schedule and ethnicity do. It is true that being basic demographic features are the ones that are better evaluated among the other determinants, but the final level of impact on sleep is no different from the others, these a priori differentiation should not be made.

Response: Defining the demographics as confounders was done by many of the reviews studied, and not initiated by the authors of the umbrella review. Though these demographic determinants are highly relevant for sleep, they are not the intended targets of sleep interventions. This approach fits the objective of the umbrella review to identify starting points for improving sleep in health practice. We agree with the reviewer that the current wording gives the impression that it was defined as such by the authors of the umbrella review. We therefore incorporated the following text in the revised manuscript: ‘A first general remark on the synthesis is that many reviews identify several generic confounders related to sleep like age, sex, chronotype, BMI or weight’ (page 12, line 297).

Comment: (1.6) The authors decided to include pain into the evaluation as a determinant of sleep in healthy people. However, pain is already a “unhealthy condition”, and cannot be discriminated or disentangled from the other diseases or illnesses. By trying to evaluate pain per se without the contribution of other morbid conditions, the authors underlined how pain does not affect sleep. This, in my opinion, leads to an incorrect assumption. The authors should either exclude it from the evaluation or decline it into the different conditions (eg, osteoarticular, neuropathic pain, etc…). In this case overgeneralization could have been detrimental.

Response: We agree with the reviewer that pain can best be classified as an unhealthy condition to make it consistent with our inclusion and exclusion criteria. We therefore removed the determinant pain from the umbrella review and added pain in the exclusion criteria: ‘The focus on healthy natural sleep led to the exclusion of reviews on a specific disease population, illness-oriented reviews on sleep (e.g. sleep and diabetes, asthma, eating disorders, HIV, or AIDS), pain (as signal of an unhealthy condition) …’ (page 8, line 197).

Comment: (1.7) In the Discussion section the authors state that definition of sleep parameters may be inconsistent and that better efforts should be made into definition of sleep parameters. This point should be better tackled. Objective sleep parameters are defined, and no inconsistencies are present. The perceived inconsistencies may result in subjective evaluation, that may not find a solution as they disperse in the complex interindividual and intraindividual variabilities of perceived events. Sleep is a complex and relevant part of our lives and the inability to generalize its subjective aspects should not be taken as our lack of methodologies, but rather as a inevitable matter to deal with. To an extent we could say that generalizing sleep would be on the same level as generalizing individual determinants of daytime activities. When evaluating such complex issues as sleep determinants, and trying to analyze each determinant per se, the drive towards generalization should give way to the knowledge that complexity and its underneath interactions are just to take into account.

Response: We agree with the reviewer that dealing with sleep involves complexity and oversimplification lies in wait when focusing solely on individual determinants. We added the following passage in the discussion section of our revised manuscript to address this properly: ‘To put in summary, sleep is a highly complex phenomenon and when studying sleep determinants, we should value its complexity instead of limiting interpretations to unambiguous definitions of sleep parameters or determinants.’ (page 34, line 795).

We also agree with the reviewer that consensus has been reached on the definition of objective sleep parameters. However, not all identified reviews use these definitions or do not even present a definition of the parameters used. We kindly refer the reviewer to our comments on reviewer 3 (comment 3.4) in which we add a remark on the interrelatedness of the determinants: ‘An important area of concern and a limitation is that determinants are interrelated, which means that the relationship between determinants and sleep outcomes is not strictly linear, but most likely more complex.’ (page 37, line 879).

Comment: (1.8) It is incorrect to define sleep as an “outcome of a range of autonomic processes and deliberate actions prior to sleep itself”. Sleep is a active process, and not the passive bystander and consequence of daytime activities and “determinants”. Sleep can directly affect daytime activities and its regulation go beyond the autonomic processes undergoing its phases. The relation sleep-daytime life is to be considered as biunivocal and at the same level, not with one as a byproduct of the other.

Response: While we definitely agree with the reviewer that the relation sleep-daytime is bidirectional and sleep is an active process, we feel that sleep itself is not merely a conscious process directed by the individual. We also feel that both are not mutually exclusive: all behavior prior to sleep itself interacts with sleep as well as processes and interferences which take place after sleep onset. Therefore, we’ve clarified this as follows in the manuscript: ‘Sleep itself is not an example of conscious behavior. Though sleep is an active process and interlinked with daytime activities, it is rather the outcome of a range of autonomic processes and deliberate actions prior to sleep itself’ (page 35, line 821).

Comment: (1.9) In the case of “disrupted sleep”, sleep hygiene is the first line approach to be taken into consideration. Sleep hygiene rules are simple, but effective, because they are the very bases, the foundation of good sleep. The fact that sleep is complex with many determinants is not a contradiction, while the authors seem to point out so. To make a parallelism with diabetes, it is true that diabetes may have various complex determinants, but if you do not follow the very simple rule of moderating sugar intake your diabetes will worsen. That is the same. Moreover, the fact sleep hygiene is not always effective should not be a reason for discouraging. For example, for primary insomnia, if the patient does not follow these rules and does not really accept that his/her will to change the situation is their primary determinant for the condition, a solution could not be found. That would be equal to saying that we should not always suggest some drugs because of some patients scarce compliance.

Response: While we agree with the reviewer that simple sleep hygiene rules are the basis for sleep improvements, we think that the sleep hygiene rules can be improved by adding relevant sleep determinants in line with reference 128. Our intention is not to discourage the use of sleep hygiene rules but to extend these rules and enhance their potential effectivity. We clarified this as follows in the manuscript: ‘In conclusion, extending the sleep hygiene rules with the identified sleep determinants, the related sleep parameters and important moderators from this umbrella review could help counsellors and their clients to become aware of their personal set of relevant determinants and identify promising starting points for sleep improvements’ (page 36, line 862).

Comment: (1.10) As a final remark and linked to the matter of complexity-simplification, the very strict criteria which finally cut-out the majority of evidence, actually oversimplified the matter. The pruning of evidence was to an extent, excessive, and the final corpora resulted impoverished rather than better specified. This was probably exemplified by the fact that some determinants actually had only one review they were taken from, which may be a source of bias for the umbrella review.

Response: We agree that there is a risk of bias due to the limitation to published reviews. The umbrella review can merely be seen as a starting point for further research as referred to in our manuscript: ‘However, this comprehensive umbrella review, covering a wide variety of determinants, provides an overarching starting point that leaves room for add-ons. Selecting the sleep determinants from published reviews results in well substantiated factors that influence sleep. On the other hand, this umbrella review does not reflect a balanced overview of all determinants since there will be a lack of reviews and meta-analyses on determinants of sleep that have been less subject to research.... ‘ (page 36, line 869)… and further on: ‘A major limitation is the fact that only part of the reviews was of sufficient quality. Additional research on sleep and sleep determinants will provide more insights in addition to those provided in the present review.’ (page 37, line 878).

In addition, we added a sub-section ‘Recommendations for future research’ in the discussion section where we address this issue: ‘This umbrella review can be merely seen as a starting point and it reflects the current availability of reviews on the determinants of sleep in natural circumstances’ (page 39, line 932).

REVIEWER 2

General comment: The manuscript describes an umbrella review aimed at providing an overview of the current evidence on determinants of natural adult sleep. The authors conducted a literature search on six electronic databases (PubMed, WoS, Embase, CINAHL, PsycInfo and Cochrane), used a shared coding system for assessing quality of the selected articles (AMSTAR2 tool) and registered the review on PROSPERO. Ninety 3 reviews and meta-analyses were identified. Results evidenced that each determinant was found to affect different sleep parameters and the relationship with sleep is influenced by both generic and specific moderators.

The topic is worth to be studies, however several pitfalls should be acknowledged.

Response: Thank you for your valuable feedback and for emphasizing the added value of studying this topic.

Comment: (2.1) The authors start their introduction traying a definition of sleep. This is a tall order issue. In 2007 Chokroverty discussed in detail the history of sleep research, evidencing that starting from the 40ths and 50ths of the last century to his present, research on the neuro-psychophysiology of sleep has advanced greatly but 2 basic questions were still open: What is sleep? and Why do we sleep? These questions are still almost open and each researcher that addresses sleep issues, usually skips its definition, preferring to adopt a descriptive approach and thus addressing its characteristics or structure or “determinants”.

Instead the authors chose to start their introduction giving a questionable definition: sleep is “a reversible and repetitive condition of diminished consciousness” using as reference Laar et al 2021 (Laar Mvd, Hadden B. Slapen als een oermens : wat de evolutie ons leert over een goede slaap. Eerste druk. ed. [Eindhoven]: Merijn van de Laar; 2021) that is in a language not accessible to all international readers. 

I suggest to give a descriptive definition (see for instance Hirshkowitz, 2004 “Sleep can be defined many ways; however, the basic core concepts remain the same. First and foremost is that sleep is a brain process. The body rests but the brain sleeps. This is not to say the body does not require sleep; there are essential body processes that occur only when the brain is asleep. Nonetheless, the brain is what does the sleeping. The second core

concept is that sleep is not a unitary phenomenon.”).

Response: We agree with the reviewer that the reference used reflects a specific perspective on sleep and is not accessible for an international audience due to language constraints. We therefore kindly accept the proposed reference instead and incorporated the following description of sleep: ‘Multiple definitions of sleep are available though several core elements of sleep are generally accepted: sleep is a brain process (while the body rests the brain sleeps), sleep is not a unitary phenomenon (it exists of different types of sleep each with their own characteristics, functions and regulatory systems) and some sleep processes are active and involve significant brain activity’ (page 2, line 40). 

Comment: (2.2) Reference 11 is very old. Many new papers could be cited for acknowledging updated effects of sleep deprivation on executive functions, mood, autonomic function, immune system, job performance and risk of traffic or industrial accidents. 

Response: We agree with the reviewer that citing more recent articles would be an improvement and therefore we added more recent citations instead: reference 10 (Irwin, 2015) and reference 11(Krause, 2017)(page 3, line 70).

Comment: (2.3) Moreover there is a consensus report about how much sleep do we need that could be cited (Watson NF, Badr MS, Belenky G, Bliwise DL, Buxton OM, Buysse D, Dinges DF, Gangwisch J, Grandner MA, Kushida C, Malhotra RK, Martin JL, Patel SR, Quan SF, Tasali E. Recommended amount of sleep for a healthy adult: a joint consensus statement of the American Academy of Sleep Medicine and Sleep Research Society. SLEEP 2015;38(6):843–844).

Response: We thank the reviewer for pointing out the scientific consensus on sleep duration. We thankfully added a remark on the consensus with the accompanying reference in our revised manuscript (page 5, line 149).

Comment: (2.4) There is a logical leap, at the end of the introduction, where the authors simply state that “Sleep is typically conceptualized broadly, containing different aspects when operationalized, so called sleep parameters (e.g., sleep duration, sleep quality, sleep disturbances or even some aspects of dreaming). As a result, sleep has a broad range of determinants”. This leap introduces the determinants of sleep that are the focus of the review while all the introduction focuses on the importance of sleep. I suggest to shorten the previous part of the introduction and to give more space to discussing why sleep may be operationalized through its different parameters and what does each parameter means. These aspects are addressed in the method section but actually they have a theoretical background that may be introduced in the introduction section.

Response: We agree with the reviewer that sleep determinants are the central theme of the umbrella review. However, we feel a need to address the importance and complexity of sleep, in order to introduce the determinants of sleep in the proper context. In line with the reviewer’s comment, we added a brief introduction of the sleep parameters in the introduction section to better balance the content of the introduction to the rest of the manuscript.

‘Sleep is a complex phenomenon and typically conceptualized broadly, containing different aspects when operationalized, so called sleep parameters (e.g., sleep duration, sleep quality, sleep disturbances or even some aspects of dreaming). For example, sleep duration is a measure for the length of sleep and sleep onset latency a measure for the time needed to fall asleep after turning off the lights. In the method section definitions of the relevant sleep parameters are provided. Different sleep measures shed a different light on experiencing sleep and are subject to different determinants of sleep.’ (page 4, line 111).

Comment: (2.5) Results section. “Socio-economic determinants consist of social and economic influences on sleep: in other words, the conditions in which people are born, grow, live work and age that reflect on their sleep. The socio-economic determinants not only include the social-economic situation of an individual but also include individual and personality related social concepts such as psychological dispositions”.

Considering personality, attachment style, sexual orientation and psychological dispositions in general as socio-economical determinants is questionable and the authors do not argument on it nor report on which bases they made that decision. It is my opinion that psychological characteristics should be distinguished from socio-economic characteristics. Moreover, the reference the authors cite for supporting their categorization (i.e. Health NSWDo. Public Health Classifications Project–Determinants of Health. Phase 2 Two: Final Report. 2010) does not include psychological characteristics/dispositions within the socio-economic determinants of health.

Response: We agree with the reviewer that the term ‘socio-economic’ in general does not refer to personal psychological characteristics. However, the Public Health Classifications Project–Determinants of Health defines social determinants as determinants that influence health, including social, cultural, and gendered roles, religious belief or spirituality, health cognition; and other societal contributors such as social attitudes, class/ caste systems, community involvement and social and support systems. We belief that this definition leaves room for adding psychological aspects to this category. In addition, the other categories of determinants of health (behavioral, biological and environmental) are less appropriate options. In order to clarify our decision, we added the following explanation: ‘Though in general the term socio-economic does not include personal psychological characteristics, the elaboration of this category within the health classification of determinants (e.g. social, cultural, and gendered roles, religious belief or spirituality, health cognition; and other societal contributors such as social attitudes, class/caste systems, community involvement and social and support systems) leaves room for including psychological-related determinants and makes it the most appropriate category within this classification (page 25 line 621).

REVIEWER 3

General comment: I must say I have quite appreciated this umbrella review aimed to summarize evidence from meta analyses and systematic reviews on the impact of the main determinants of natural sleep

In fact, I would like credit quite many aspects: good organization, clear explanation of methods, rigorous approach to data search, well-balanced theoretical comments in the discussion including useful suggestions and rules of thumbs for applicative/clinical purposes.

Still, there are a few major issues, plus some minor points that could be addressed and improved throughout the paper, that I would definitely encourage the authors to reconsider in light of the comments that I am adding below:

Response: Thank you for your kind words and compliments. We much appreciate your positive criticism and feel that the manuscript truly benefits from the improvements resulting from it.

MAJOR ISSUES

Comment (3.1): First of all, I suggest that some cautionary comments are made on the intrinsic limit of the “umbrella review” approach, that unavoidably suffer from a literature bias. What I mean is that some determinants will be overestimated because more reviewed in the literature whereas others, often even more important, will be neglected due to the partial/total lack of reviews (not necessarily corresponding to a lack of experimental research). More specifically, in this article there are some extremely relevant determinants of sleep which have received very little space (e.g., cognitive activity, gender) as compared to rather minor determinants (meditational physical activity, music, intermittent fasting), and some other are completely absent (e.g., electromagnetic fields are among the most debated environmental determinants, and bedding habits, such as posture, mattresses etcetera, are also not mentioned).

Response: We definitely agree with the reviewer that there is a strong risk of bias due to the availability of reviews on this topic. We therefore added the following limitation to the discussion section: ‘Selecting the sleep determinants from published reviews results in well substantiated factors that influence sleep. On the other hand, this umbrella review does not reflect a balanced overview of all determinants since there will be a lack of reviews and meta-analyses on determinants of sleep that have been less subject to research.’ (page 37, line 871).

In addition, we added a sub-section ‘Recommendations for future research’ in the discussion section where we address this issue: ‘This umbrella review can be merely seen as a starting point and it reflects the current availability of reviews on the determinants of sleep in natural circumstances. Our recommendations for future research are;

• More systematic reviews or meta-analyses on determinants that receive less attention, for example bedding and mattresses or cognitive activity, would greatly enhance the quality and usability of the overview (page 39, line 932).

Comment: (3.2) At this regard, I wonder whether the paragraph on age does really cover all the reviews published on the topic. By heart, I remember at least two that are pioneering and extremely important: Bliwise DL, Sleep 1993, 16(1):40-81; Ohayon MM et al., Sleep, 2004;27(7):1255–1273. What is the reason for their exclusion?

Response: Thank you for your remark. We checked the two articles mentioned and they were indeed included in the original download. Regarding their in-/exclusion:

• Bliwise DL, Sleep 1993, 16(1):40-81 : This article is written by one author only which makes it not in line with the Cochrane guidelines for systematic reviews.

• Ohayon MM et al., Sleep, 2004;27(7):1255–1273 (title: Meta-analysis of quantitative sleep parameters from childhood to old age in healthy individuals: developing normative sleep values across the human lifespan); The focus of this article is on childhood and adolescents. Nevertheless, it does contain results on (older) adults in line with the results displayed in the included articles. 

We added a remark on this issue in our revised manuscript: ‘The exclusion of reviews on children and adolescents, led to less attention for research on some determinants, such as social media use, which was mainly targeted at children and adolescents [133]. Similarly, research on age mostly included and focused on children and adolescents [134](page 37, line 875).

In addition, we ran an additional double check on age and sleep but all other articles in the downloads are justly excluded for the following reasons:

• No systematic review according to the Cochrane criteria used

• Diseased population 

• Article on children or adolescents 

Comment: (3.3) I am also kind of surprised by the assessment of the analyzed reviews’ quality (Page 10 and table 2). Although the method used is appropriate and well-described, I am negatively impressed by the very low proportion of determinants supported by “sufficient quality” literature (5 out of 29, namely 0 out of 3 for biological, 2 out of 11 for behavioral, 1 out of 6 for environmental, 2 out of 9 for socio-cultural). In my opinion, the rating could be too conservative, given that RoB is very seldom addressed even in allegedly excellent reviews that provide a precious contribution (I know some of them myself among the cited papers). Therefore, one option would be to change the “sufficient, mediocre, questionable” scale in “good, sufficient, questionable”. Otherwise, it would be necessary to report the general low quality of the available literature, in the Discussion, as a major limitation of the study.

Response: We agree with the reviewer that the number of reviews of sufficient quality, measured against the AMSTAR-2 criteria is low and should be well addressed as a major limitation of the reviews on the topic of sleep. Therefore, we emphasized this more strongly in our revised manuscript: ‘A major limitation is the fact that only part of the reviews was of sufficient quality.’ (page 37, line 882).

Comment: (3.4) I think the authors should try to be somehow clearer on the definition of sleep quality, which is an extremely tricky matter. First, objective sleep measures traditionally considered as indices of sleep quality (such as, for example, sleep depth and sleep continuity), do not utterly correspond to the determinants of subjective sleep perception, including aspects such as feeling refreshed at awakening. So the idea of collapsing in a single measure both objective and subjective sleep quality is questionable. 

Secondly, there is a close relationship between different determinants of sleep quality. Clearly, I realize that some sort of generalization is needed to compare different data sources and to get an overall view, but the theoretical problem should be addressed in more details and represents a limitation that should be definitely mentioned.

Response: We fully agree with the reviewer that defining sleep quality is a tricky matter. As this comment has been made by reviewer 1 as well, we kindly refer to our response on comment (1.1). In addition, we added a comment on the interrelatedness of sleep quality parameters in our discussion section: ‘An important area of concern and a limitation is that determinants are interrelated, which means that the relationship between determinants and sleep outcomes is not strictly linear, but most likely more complex.’ (page 37, line 879).

Comment: (3.5) Also, talking of interconnections between variables, I have not fully understood how the data have been treated concerning interdependent determinants. For example, how can we disentangle what is said about the effects of chronotypologies from the well-known evidence that they show consistent age-related changes? What about the possibility that socio-economic status is reflected from varying living conditions (i.e., from modifications of the environmental determinants)? I suppose some explanation is provided at page 12, lines 252-254, but it frankly seems very unclear to me.

Response: 

We agree with the reviewer that there is still a lot to investigate regarding sleep determinants and their interdependency. As this goes beyond the objective of our umbrella review, we added this valuable comment to the recommendations for further research (page 39, line 929). In addition, we added a remark on the interdependency of the determinants in the discussion section in line with your previous comment (3.4)‘ (page 37, line 879).

Comment: (3.6) Physical activity and sedentary behavior are two sides of the same medal. Also, video gaming is not a specific type of sedentary behavior. Therefore, I would definitely combine them in the same paragraph, keeping social media and video games separated.

Response: We agree with the reviewer that sedentary behavior and physical activity are interrelated. However, in research on physical activity, sedentary behavior is seen as separate behavior since sedentary behavior cannot simply be compensated by physical activity (see for example Berninger (2020) and Van der Ploeg (2017)). 

The rationale to combine sedentary behavior, gaming and social media use is the fact that the latter two in general are a subset of sedentary behaviors. We agree with the reviewer that it would be clearer and more consistent to keep sedentary behavior distinct from gaming and social media use and therefore we split both the text and table into sedentary behavior versus (video)gaming and social media use. (page 18, line 485).

Comment: (3.7) Finally, there is some sort of confusion on the notion of “natural sleep”. In the Inclusion and Exclusion criteria section, the authors state (page 7, lines 148-151) that “the focus on healthy natural sleep led to the exclusion of reviews on a specific disease population, illness-oriented reviews on sleep (e.g. sleep and diabetes, asthma, eating disorders, HIV, or AIDS), sleep diseases (e.g. bruxism, narcolepsy, sleep apnea) or on excess behavior and addictions (e.g. problematic smart phone use, alcoholism)”. I can agree on this choice, but this is somehow contradictory with the insertion of “pain” as a biological determinant and, above all (page 11, lines 239-242), of sleep medication as a mediator (since sleep medications exclude the notion of natural sleep by definition).

Response: A valid point from the reviewer. As the remark on including pain has also been made by reviewer 1, we kindly refer to comment 1.6 and the accompanying response. 

Regarding sleep medication, as mediator of sleep in natural circumstances, we also agree with the reviewer and removed the passage concerning sleep medication ‘Sleep medication is a generic mediator for all determinants on sleep.’ (page 12, line 305).

MINOR ISSUES

Comment: (3.8) The notion of “sleep determinants” could not be clear to all readers. Please, give a short definition already in the abstract (page 1, line 15)

Response: We added a short definition of sleep determinants in the abstract: ‘Research on sleep determinants, factors that positively or negatively influence sleep, is fragmented.’ (page 1, line 15).

Comment: (3.9) The “results” section in the abstract is too generic. Please, identify and mention either the most impacting determinants or the most important variables that are affected.

Response: Though we agree with the reviewer that the “Results” section in the abstract is rather generic we feel that it is impossible to highlight certain determinants since the impact and importance of the determinants varies per individual per situation. However, we feel that a clarification would surely improve the abstract and therefore we incorporated the following in the Abstract: ‘In total 93 reviews and meta-analyses were screened, resulting in a total of 30 identified determinants. The impact of each determinant differs per individual and per situation. Each determinant was found to affect different sleep parameters and the relationship with sleep is influenced by both generic and specific moderators.’ (Page 1, line 25).

Comment: (3.10) 

Page 2, line 30: a full stop point is missing between “review” and “Extending”

Page 2, line 39: Either remove the full stop point between [2] and “Whereas" or replace “whereas” with “Instead,”

Page 2, line 47: Replace “less “ with “decreased” or “reduced” and replace “is” with “are”

Response: Thank you for your thorough check-up! We corrected al these mistakes.

Comment: (3.11) Page 2, line 47: Is there really a “causal” link between sleep disturbances and diabete? In the abstract of the quoted paper (Reutrakul and Van Cauter, 2014) it is stated that “Several large prospective studies suggest that these sleep disturbances ARE ASSOCIATED with an increased risk of incident diabetes”. Please, verify.

Response: Indeed, the article does not conclude on a causal link between sleep disturbances and diabetes but between sleep disturbances and abnormal glucose metabolism, which might result in diabetes. That is the reason why we used exactly this remark in our manuscript: ‘Disturbances of sleep, as indicated by – for example – reduced sleep duration or sleep quality, are causally linked to abnormal glucose metabolism, which might result in diabetes type 2 or metabolic syndrome [5].’ (page 3, line 51).

Comment: (3.12) Page 3, line 64: “ (…) sleep duration of 90% of Dutch adults is in accordance with the recommendations of the American Association of Sleep Medicine (AASM)”. Please, specify what is this recommendation about sleep duration.

Response: We added the recommendation in line with the reviewer’s comment and changed the reference: ‘Longitudinal research shows that sleep duration of 90% of Dutch adults is seven to nine hours, in accordance with the recommendations of the National Sleep Foundation (NSF), …’ (page 3, line 77).

Comment: (3.13) Page 3, lines 67-69: “A poll (…) enough time to sleep [15].” Confused sentence, please rephrase.

Response: We rephrased the sentence to: ‘A survey by the US National Sleep Foundation (NSF) in 2020 reported that 16% of US adults did not feel sleepy during a typical week. Of all the respondents that did report feeling sleepy, 55% indicated not sleeping well enough compared to 44% not having enough time to sleep [15].’ (page 3, line 82).

Comment: (3.14) Page 3, line 72: “The results (…) United Kingdom [15]” Are these all surveys on habitual sleep?

Response: The surveys were done on a random sample of adults (aged 25 to 55 years old) comparing sleep times, attitudes, habits and bedtime routines of those in the United States, Canada, Mexico, the United Kingdom, Germany and Japan. (page 4, line 106).

Comment: (3.15)

• Page 4, line 75: Please, erase “one’s”

• Page 4, line 82: please, replace “with” with “according to”

• Page 6, Table 1: right column, second row (SOL): erase “with which”

• Page 11, line 225: please, replace “around before mentioned” with “around the before mentioned”

• Page 11, line 233: please, replace the full stop after weight with a comma.

• Page 12, line 246: please, replace “mental” with “psychological”

Response: Thank you again! We most appreciate the effort you have taken to improve our manuscript. We incorporated all changes.

Comment: (3.16) Page 4, line 78: “sleep disturbances” and “aspects of dreaming” are not sleep parameters.

Response: We agree with the reviewer that dreaming is in general not considered a valid sleep parameter. The point we want to make here is that some researchers take the definition of a parameter rather literally and do consider aspects of dreaming as a sleep parameter. Whereas for aspects of dreaming this is occasionally done, sleep disturbances are more often considered important sleep parameters for defining sleep quality [ 28, 29]. Though consensus has been reached on the definition of sleep parameters, researchers still use other definitions and parameters to describe the characteristics of sleep. To clarify this, we added the following to our revised manuscript: ‘In the method section definitions of the sleep parameters, that are generally accepted in scientific research, are provided’ (page 4, line 115).

Comment: (3.17) Page 10, lines 218-220: “We could not (…) determinants”. Unclear sentence, please rephrase.

Response: We rephrased the sentence to: ‘We could not use the classification proposed in this scoping review since it covered only part of our (categories of) determinants.’ (page 11, line 280).

Comment: (3.18) Page 11, line 226: what does “the nature of the relation with sleep” mean?

Response: With the nature of the relation, we meant the characteristics of the relation, for example a positive or negative direction of the relation. We rephrased the sentence to: ‘A narrative synthesis of the findings will be provided, structured around the before mentioned categorization of the determinants, the relation with sleep and the specific sleep parameters that are impacted.’ (page 11, line 287).

Comment: (3.19) Page 12, lines 249-250: "Biological determinants (…) for an individual”. Incorrect, see for instance age and chronotypologies

Response: We acknowledge the incorrect formulation of the biological determinants. What we meant to say is that these determinants change over time but that it is impossible to control their impact on sleep. We therefore incorporated the following text lines into our revised manuscript: ‘It is practically impossible for an individual to change one’s biological determinants of sleep, but they follow the rules of nature and as such can explain sleep differences between individuals and between different life phases.’ (page 12, line 314).

Comment: (3.20) Page 12, “Age and sex” section: the age range of the umbrella review should be specified, either here or in the Method. What is the younger and the older age included?

Response: Unfortunately, the age ranges differ per review, but we agree that a specification of age ranges is needed. Therefore, we added these ranges to our revised manuscript: ‘The term elderly may refer to different ages but is generalized to ages 58 or 60 and above in the included studies.’ (page 13, line 339).

Comment: (3.21) Page 12, lines 263-264: “However (…) men.” Good point! This distinction between subjective and objective sleep quality should be kept in mind also elsewhere (see major issues, comment n. 4).

Response: We kindly refer to our response on comment 3.4 and 1.1.

Comment: (3.22) Page 12, lines 265-266: “Elderly (…) deep sleep.” In the comparison with men? Or with other ages? Or relative to other stages? Please, specify.

Response: We agree with the reviewer that the comparison is not clearly identified and therefore we incorporated the following change in the revised manuscript: ‘Comparing elderly men to women: elderly men tend to have more light sleep and elderly women more deep sleep.’ (page 13, line 340).

Comment: (3.23)

• Page 12, line 266: Please, replace “on” with “of”

• Page 13: please, add “about” between “when” and “individual” and replace “or when they are most alert” with “and when they prefer to stay awake”

Response: Many thanks! We incorporated all changes.

Comment: (3.24) Page 13, line 277: is it “subjective” sleep quality?

Response: Thank you for pointing out this unclarity. Indeed, we refer to subjective sleep quality and therefore we added the term subjective: ‘E-type, compared to I-type or M-type, shows more often a decrease in sleep time, decreased subjective sleep quality and less sleep efficiency (the latter when compared to M-type only).’ (page 13, line 352).

Comment: (3.25) Page 15, line 307: what kind of “sleep disruptions”?

Response: The review mentions sleep disruption without further details and defines WASO as an indicator of sleep continuity or sleep disruption. We therefore incorporated the following in our revised manuscript: ‘On the other hand, the higher the dose, the more often sleep disruption (WASO) occurs and the lower the percentage of REM-sleep.’ (page 15, line 393).

Comment: (3.26) Page 16, lines 334-335: “For older (…) impact on sleep”. Unclear please rephrase

Response: We rephrased this passage to: ‘Whereas most studies show that a diet rich of these products facilitate sleep of good quality, for older adults a similar but negative relation has been found: less milk and less fish may have a negative impact on sleep.’ (page 16, line 421).

Comment: (3.27) Page 20, lines 424-425: “(Physical) (…) change in sleep”. I do not understand the meaning and implication of this sentence.

Response: Though we used the definition of environmental health determinants and slightly adapted it to sleep determinants, the definition may not only cover involuntary but also purposeful changes in the environment. Therefore, we slightly refined the definition to: ‘(Physical) environmental determinants exist of any external factor (biological, chemical, physical) that can be linked to a change in sleep.’ (page 21, line 533).

Comment: (3.28) Page 33, line 666: “on determinants”. Please, specify “on those determinants that have been previously covered by adequate systematic reviews”

Response: Thank you for the specification. We changed the sentence to: ‘The present umbrella review provides an overview of the current evidence on those determinants of natural adult sleep that have been previously covered by adequate systematic reviews.’ (page 33, line 784).

Comment: (3.29) Page 33, line 674: “longevity” ???

Response: Thank you for pointing out this incorrect translation. What we meant to say is that one dimension of physical exercise is about exercising only once versus sustained exercising over a long period of time. We incorporated the following text into our revised manuscript ‘.. e.g. physical exercise has the dimensions type, duration, intensity and regularity of training. (page 34, line 792).

Comment: (3.30) Page 34, lines 690-691: Unclear sentence, please rephrase.

Response: We rephrased the passage to: ’The night is a reflection of the day, meaning that daytime behavior and daytime experiences impact sleep by so called somnoprints. A somnoprint is a characteristic sleep pattern, related to events occurring during waking.’ (page 35, line 819).

Comment: (3.31) Page 34, lines 704-708: Excellent point!

Response: Thank you!

Reviewer 4

General comment: I read with interest the manuscript entitled “Determinants of natural adult sleep: an umbrella review.” In this study Philippens and co-authors aimed at providing an overview of the determinants of natural sleep by conducting an “umbrella review” i.e., a meta-review based on previously published reviews and meta-analyses. Their final goal was to provides a practical, scientifically-based, background to develop novel interventions aimed at improving sleep quality.

The authors identified and analyzed 93 reviews and meta-analyses. Results were categorized in four main categories: biological, behavioral, environmental, and socio-economical determinants. Quality of the selected articles was assessed using a method based on the AMSTAR2 tool.

Overall, the authors concluded that each determinant affect different sleep parameters although with a high degree of overlap.

The study aim, while relevant, is extremely ambitious, far too ambitious to be addressed in a single manuscript. The research question is too broad, and the authors failed to provide a comprehensive nor "practical" picture for any of the specific determinant of sleep under investigations.

Response: Thank you for mentioning the relevance of our umbrella review. As far as the ambition of this study goes, we feel that this umbrella review is a first but necessary step to map the determinants of sleep. Only thereafter the overview can and should be extended with additional determinants, interdependencies and their effects on different sleep parameters. 

Comment: (4.1) The topics covered are far too broad. This issue has an important influence on the degree of analytical depth with which the specific determinants are analyzed. Indeed, the results described are very academic and does not add anything new to the knowledge already possessed by clinicians and health professionals working in the field of sleep. 

I strongly suggest narrowing down the review to a maximum two determinant: biological and behavioral determinants of sleep are the most relevant topic to cover with the aim of developing novel interventions on the other hand environmental, and socio-economical determinants are far less studies and could be more interesting to review.

Response: While we agree with the reviewer that the scope of this umbrella review is broad, we feel that our chosen approach best fits the objective of the umbrella review. We aim at providing a broad perspective on the sleep determinants: providing both experts and non-experts on the topic of sleep a practical overview with relevant sleep determinants when dealing with patients experiencing sleep issues. This is especially relevant since the PROSPERO database currently shows that there is no similar umbrella review pending providing one single overview of determinants on sleep. In addition, this umbrella review could serve as a starting point for future research, a framework, to get a grip on the complexity of sleep and sleep behavior that then needs to be further explored in more in-depth studies. For many early career researchers or those new to the field, this might be very valuable to get an overview of the complexity before diving deeper into the topic. 

Comment: (4.2) The review was not performed according to PRISMA guidelines.

While this specific type of review may not be covered by the PRISMA guidelines, it would have helped the authors define, and subsequently analyze, a more precise research question.

Response: We agree with the reviewer that the PRISMA guidelines, though not entirely matching the approach taken for an umbrella review, may be beneficial. Please note that we followed the guidelines whenever appropriate.

Comment: (4.3) The manuscript its present version, runs the risk of being a sort of "shopping list" of available results. The authors are strongly encouraged discuss results more proactively by adding a few lines of reasoning that help a general recap. The Discussion section could be enriched by discussing more explicitly and extensively some theoretical aspects.

Response: We agree with the reviewer that a theoretical elaboration of sleep would provide even more in-depth information. However, this is not the objective of this umbrella review. We used the overall overview by means of the framework to show that in some areas and for some specific determinants the evidence is extensive whereas other areas leave a lot of room for additions. We added this valuable suggestion to our recommendations for future research (page 39, line 943). 

Comment: (4.4) The manuscript would really benefit from a “Research Agenda/Future Prospective” section.

Response: Thank you for this excellent suggestion. The recommendations for future research are partly presented in the discussion section. We added a subheader ‘Recommendations for future research’, moved all recommendations to this additional section and made additions in line with the reviewers’ comments. (page 39, line 932).

---

## [Decision Letter · Decision Letter 1]

26 Jul 2022

PONE-D-22-10437R1Determinants of natural adult sleep: an umbrella reviewPLOS ONE

Dear Dr. Philippens,

Thank you for submitting your manuscript to PLOS ONE. After careful consideration, we feel that it has merit but does not fully meet PLOS ONE’s publication criteria as it currently stands. Therefore, we invite you to submit a revised version of the manuscript that addresses the points raised during the review process.

In particular, the Reviewers and I appreciated your revision of your manuscript. However, a few points originally raised by the Reviewers have not been dealt with satisfactorily. I therefore encourage you to further revise your manuscript incorporating all the changes originally suggested by the reviewers and remarked in their last comments, which are detailed below.

We look forward to receiving your revised manuscript.

Kind regards,

Alessandro Silvani, M.D., Ph.D.

Academic Editor

PLOS ONE

Journal Requirements:

Reviewers' comments:

Reviewer's Responses to Questions

**Comments to the Author**

1. If the authors have adequately addressed your comments raised in a previous round of review and you feel that this manuscript is now acceptable for publication, you may indicate that here to bypass the “Comments to the Author” section, enter your conflict of interest statement in the “Confidential to Editor” section, and submit your "Accept" recommendation.

Reviewer #1: (No Response)

Reviewer #2: All comments have been addressed

Reviewer #3: All comments have been addressed

Reviewer #4: (No Response)

2. Is the manuscript technically sound, and do the data support the conclusions?

Reviewer #1: Yes

Reviewer #2: Partly

Reviewer #3: Yes

Reviewer #4: Partly

3. Has the statistical analysis been performed appropriately and rigorously? 

Reviewer #1: Yes

Reviewer #2: N/A

Reviewer #3: Yes

Reviewer #4: N/A

4. Have the authors made all data underlying the findings in their manuscript fully available?

Reviewer #1: Yes

Reviewer #2: Yes

Reviewer #3: Yes

Reviewer #4: No

5. Is the manuscript presented in an intelligible fashion and written in standard English?

Reviewer #1: Yes

Reviewer #2: Yes

Reviewer #3: Yes

Reviewer #4: Yes

6. Review Comments to the Author

Reviewer #1: I read with pleasure the thorough revision made by the authors to their paper “Determinants of natural adult sleep: an umbrella review”. From my part the authors have addressed almost all the points, there are only some minor aspects still requiring clarification or to be remarked:

- The authors state that they did not exclude reviews based on the registration method of sleep. I suggest clarifying the following sentence as it may be unclear to what extent the “referring to” was considered as an exclusion parameter (page 8 lines 169-172) “Since we wanted to identify determinants of natural sleep, we excluded reviews 170 referring to sleep regulation (e.g. endocrine system), symptoms or consequences of sleep or 171 sleep deprivation (for example sleepiness or diminished cognitive performance due to sleep 172 deprivation) or registration methods of sleep parameters (e.g. actigraphy, EEG, enquiries).”

- I agree with the definition differentiating determinants and interventions. However, on the practical side it really does not change a lot. For example, when considering meditative activity many of the reviews-metanalyses cited described randomized controlled trials for these activities where the participants were asked how these activities impacted many aspects of their lives, including sleep. As a randomized controlled trial is not a thing a person normally does and as the participants knew these activities could influence sleep they should be considered also as intervention.

- I agree that some demographic confounders cannot be excluded as they were already included in the reviews-metanalyses, and I totally agree with the incorporated specification in the manuscript. However, it should be also pointed out that BMI is indeed target of sleep interventions, as weight control is the first line action to be suggested when addressing Obstructive Sleep Apneas.

Finally, the authors have both stated in revising their work that a) “this is a review which could be either used to support the practical use of the knowledge on sleep determinants in day-to-day practice of health professionals in a primary care setting” and b) “the review can be merely seen as a starting point for further research”. As a last suggestion, I would encourage the authors to focus on one of these goals, taking into consideration all the revisions, as again, with a single general work it could be highly difficult to tackle both points.

Reviewer #2: The authors addressed all the concerns raised by my revision ecept one. Namely, in my previous revision, I commented that:

"Considering personality, attachment style, sexual orientation and psychological dispositions in general as socio-economical determinants is questionable and the authors do not argument on it nor report on which bases they made that decision. It is my opinion that psychological characteristics should be distinguished from socio-economic characteristics. Moreover, the reference the authors cite for supporting their categorization (i.e. Health NSWDo. Public Health Classifications Project–Determinants of Health. Phase 2 Two: Final Report. 2010) does not include psychological characteristics/dispositions within the socio-economic determinants of health"

The authors' answer is:

"We agree with the reviewer that the term ‘socio-economic’ in general does not refer to personal psychological characteristics. However, the Public Health Classifications Project–Determinants of Health defines social determinants as determinants that influence health, including social, cultural, and gendered roles, religious belief or spirituality, health cognition; and other societal contributors such as social attitudes, class/ caste systems, community involvement and social and support systems. We belief that this definition leaves room for adding psychological aspects to this category. In addition, the other categories of determinants of health (behavioral, biological and environmental) are less appropriate options. In order to clarify our decision, we added the following explanation: ‘Though in general the term socio-economic does not include personal psychological characteristics, the elaboration of this category within the health classification of determinants (e.g. social, cultural, and gendered roles, religious belief or spirituality, health cognition; and other societal contributors such as social attitudes, class/caste systems, community involvement and social and support systems) leaves room for including psychological-related determinants and makes it the most appropriate category within this classification (page 25 line 621)."

I do not agree that the definition of ‘socio-economic’ determinants given by the Public Health Classifications Project–Determinants of Health leaves room to include psychological determinants within the socio-economic category. I thing that "behavioral" category is much more approriate.

Reviewer #3: Having all reviewers' comments been thoroughly addressed, the manuscript ends up being much improved relative to the previous version.

Reviewer #4: Unfortunately, the authors did not respond to the points raised satisfactorily. Specifically, I asked to discuss the findings more proactively and the authors overlooked this comment by stating that this would fall outside the aims of the review.

Secondly, I asked for a "Research Agenda/Future Perspectives" section to be added. Although the authors have included this section, the points discussed are too broad and fail in providing relevant recommendations for future research.

7. PLOS authors have the option to publish the peer review history of their article (what does this mean?). If published, this will include your full peer review and any attached files.

Reviewer #1: **Yes: **Luca Baldelli

Reviewer #2: No

Reviewer #3: **Yes: **Gianluca Ficca, MD, PhD

Reviewer #4: No

---

## [Author Response · Author response to Decision Letter 1]

28 Sep 2022

RESPONSE TO REVIEWERS

Reviewer #1

I read with pleasure the thorough revision made by the authors to their paper “Determinants of natural adult sleep: an umbrella review”. From my part the authors have addressed almost all the points, there are only some minor aspects still requiring clarification or to be remarked:

Comment 1.1: The authors state that they did not exclude reviews based on the registration method of sleep. I suggest clarifying the following sentence as it may be unclear to what extent the “referring to” was considered as an exclusion parameter (page 8 lines 169-172): “Since we wanted to identify determinants of natural sleep, we excluded reviews referring to sleep regulation (e.g. endocrine system), symptoms or consequences of sleep or sleep deprivation (for example sleepiness or diminished cognitive performance due to sleep deprivation) or registration methods of sleep parameters (e.g. actigraphy, EEG, enquiries).”

Response 1.1: Thank you and glad to hear that you mostly agree with the revisions on our manuscript. We agree with the reviewer that the terminology “referring to” is multi-interpretable and therefore we used the following wording instead: “Since we wanted to identify determinants of natural sleep, we excluded reviews that focus on sleep regulation (e.g. endocrine system), symptoms or consequences of sleep or sleep deprivation (for example sleepiness or diminished cognitive performance due to sleep deprivation) or registration methods of sleep parameters (e.g. actigraphy, EEG, enquiries). [Page 8, line 207]

Comment 1.2 I agree with the definition differentiating determinants and interventions. However, on the practical side it really does not change a lot. For example, when considering meditative activity many of the reviews-metanalyses cited described randomized controlled trials for these activities where the participants were asked how these activities impacted many aspects of their lives, including sleep. As a randomized controlled trial is not a thing a person normally does and as the participants knew these activities could influence sleep they should be considered also as intervention.

Response 1.2: We agree with the reviewer that the difference between randomized controlled trials and interventions is a thin line. An RCT provides participants with a strict regime in the experimental group, which is indeed not the same as routine behavior. The difference however, that has led to our decision to include these reviews, is that within an RCT all other factors (except the determinant) are controlled within all reasonableness. An intervention, however, typically consists of multiple components that make it hard or even impossible to single out the effect of one determinant/factor. To address this issue properly we added the following text to the Discussion section (bold section): “An important area of concern and a limitation is that determinants are interrelated, which means that the relationship between determinants and sleep outcomes is not strictly linear, but most likely more complex. This interrelatedness is one of the reasons we excluded studies on sleep interventions as it is difficult to single out the impact of one individual factor as a determinant.” [Page 37, line 905]

In addition, we added a suggestion for future research on intervention studies on sleep as a different and valuable perspective. [Page 40, line 976].

Comment 1.3: I agree that some demographic confounders cannot be excluded as they were already included in the reviews-metanalyses, and I totally agree with the incorporated specification in the manuscript. However, it should be also pointed out that BMI is indeed target of sleep interventions, as weight control is the first line action to be suggested when addressing Obstructive Sleep Apneas.

Response 1.3: We agree with the reviewer that BMI has a profound effect on sleep and weight control should be an important focus of interventions by health practitioners supporting overweight clients with sleep improvements. We therefore added the following text lines to the discussion section: “Regarding weight (or BMI) it should be noted that overweight has in many cases a direct relation with obstructive sleep apneas, thereby severely impacting sleep quality. Health practitioners supporting overweight patients should address weight control as a first line of action. [Page 36, line 868]

Comment 1.4: Finally, the authors have both stated in revising their work that a) “this is a review which could be either used to support the practical use of the knowledge on sleep determinants in day-to-day practice of health professionals in a primary care setting” and b) “the review can be merely seen as a starting point for further research”. As a last suggestion, I would encourage the authors to focus on one of these goals, taking into consideration all the revisions, as again, with a single general work it could be highly difficult to tackle both points.

Response 1.4: Our manuscript would indeed benefit from one objective only. Our main objective is to support the practical use of the knowledge on sleep determinants in day-to-day practice of health professionals in a primary care setting. We later on mention that the review can be seen as a starting point for further research. The reason behind this remark is not to add another objective but to make readers aware of the fact that revisions of this umbrella review are to be expected. 

To further clarify this focus on the practical use for health professionals we added the following text lines (we kindly refer to comment 4.2 as it is addressing this topic as well):“This umbrella review provides an overview of the current stand on research on sleep determinants, useful in day-to-day practice of health professionals addressing sleep issues with their patients. To enhance the practical use of the sleep determinants in daily practice, we recommend the presentation of the determinants in a well-structured and summarizing model, providing a subject for further research. Since sleep research is a trending topic it reflects the current availability of reviews on the determinants of sleep in natural circumstances but further research is to be expected.” [Page 39, line 956]

Reviewer #2: 

Comment 2.1; The authors addressed all the concerns raised by my revision except one. Namely, in my previous revision, I commented that:

"Considering personality, attachment style, sexual orientation and psychological dispositions in general as socio-economical determinants is questionable and the authors do not argument on it nor report on which bases they made that decision. It is my opinion that psychological characteristics should be distinguished from socio-economic characteristics. Moreover, the reference the authors cite for supporting their categorization (i.e. Health NSWDo. Public Health Classifications Project–Determinants of Health. Phase 2 Two: Final Report. 2010) does not include psychological characteristics/dispositions within the socio-economic determinants of health"

The authors' answer is:

"We agree with the reviewer that the term ‘socio-economic’ in general does not refer to personal psychological characteristics. However, the Public Health Classifications Project–Determinants of Health defines social determinants as determinants that influence health, including social, cultural, and gendered roles, religious belief or spirituality, health cognition; and other societal contributors such as social attitudes, class/ caste systems, community involvement and social and support systems. We belief that this definition leaves room for adding psychological aspects to this category. In addition, the other categories of determinants of health (behavioral, biological and environmental) are less appropriate options. In order to clarify our decision, we added the following explanation: ‘Though in general the term socio-economic does not include personal psychological characteristics, the elaboration of this category within the health classification of determinants (e.g. social, cultural, and gendered roles, religious belief or spirituality, health cognition; and other societal contributors such as social attitudes, class/caste systems, community involvement and social and support systems) leaves room for including psychological-related determinants and makes it the most appropriate category within this classification (page 25 line 621)."

I do not agree that the definition of ‘socio-economic’ determinants given by the Public Health Classifications Project–Determinants of Health leaves room to include psychological determinants within the socio-economic category. I think that "behavioral" category is much more appropriate.

Response 2.1: Thank you for pointing out the fact that the header of this section does not cover the content. We feel that psychological dispositions are the pillars/foundation on which behavior is manifested but that it is not part of actual behavior itself. Based on the reviewer’s feedback, however, we now changed the header of the section to “Personal and socio-economic determinants of sleep” [Page 25, line 625] and added a short explanation in the introduction of the paragraph (text in bold):

“Personal and socio-economic determinants consist of personal, social and economic influences on sleep: in other words, the conditions in which people are born, grow, live, work and age that reflect on their sleep. Though the health classification of determinants originally labeled this category as “socio-economic determinants”, the elaboration of this category (e.g. social, cultural, and gendered roles, religious belief or spirituality, health cognition; and other societal contributors such as social attitudes, class/caste systems, community involvement and social and support systems) leaves room for including psychological-related determinants and makes it the most appropriate category within this classification. The personal and socio-economic determinants therefore not only include the social-economic situation of an individual but also include individual and personality related social concepts such as psychological dispositions. Sleep is a fundamental biological activity but can also be perceived as a complex social process in which the characteristics of the individual interact with the social environment.”[Page 25, line 626]

Further adaptations in the manuscript (in bold):

Results were categorized in four main categories: biological, behavioral, environmental and personal/socio-economical determinants. [Page 1, line 24]

Therefore, we clustered and present the sleep determinants in four categories analogous to the classification of the Public Health Classifications Project for Determinants of Health: biological, behavioral, (physical) environmental and personal/socio-economic determinants of sleep [3]. [Page 11, line 286]

Second, some reviews combine several determinants within one of the four categories (biological, behavioral, (physical) environmental or personal/socio-economical) but none of the reviews covers determinants in more than one category. [Page 12, line 309]

Social and cultural aspects of the environment are included in the category of personal/socio-economic determinants. [Page 21, line 542]

Table 6 provides an overview of all identified personal and socio-economic determinants of sleep. [Page 31, line 787]

Title of table 6: Overview of personal and socio-economic determinants of sleep [Page 31, line 789]

Header of the first column in table 6: Personal and socio-economic determinants. [Page 31, line 789]

We found a wide variety of sleep determinants that were clustered around four main categories: biological, behavioral, environmental and personal/socio-economic determinants. [Page 34, line 798]

Personal and socio-economical determinants impact sleep directly by positive social relations and positive psychological dispositions. [Page 34, line 802]

Reviewer #3: 

Comment 3.1: Having all reviewers' comments been thoroughly addressed, the manuscript ends up being much improved relative to the previous version.

Response 3.1; Thank you for acknowledging the improvements and for your contribution to them.

Reviewer #4: 

Unfortunately, the authors did not respond to the points raised satisfactorily. 

Comment 4.1: Specifically, I asked to discuss the findings more proactively and the authors overlooked this comment by stating that this would fall outside the aims of the review.

Response 4.1: 

Thank you for pointing out the lack of a general recap and proactive discussion of the results. We agree that our manuscript could benefit from pointing out the most important findings from our perspective and therefore we now added this in our Discussion section: “We found a wide variety of sleep determinants that were clustered around four main categories: biological, behavioral, environmental and personal/socio-economic determinants. Biological determinants are practically impossible to change but they can explain differences between individuals and life phases and provide as such valuable information on natural sleep developments. Research on environmental determinants is limited to specific categories such as light and noise. Personal and socio-economical determinants impact sleep directly by positive social relations and positive psychological dispositions. Moreover, they indirectly impact sleep by increased stress levels due to social and personal circumstances. Behavioral determinants provide the most promising approach to improve sleep though it is complex as behavior in general has many parameters and the effect changes with the specifics of these parameters. [Page 34, line 797] 

Comment 4.2; Secondly, I asked for a "Research Agenda/Future Perspectives" section to be added. Although the authors have included this section, the points discussed are too broad and fail in providing relevant recommendations for future research.

Response 4.2: We agree with the reviewer that the points that are addressed in the research agenda are broad. In the text lines suggested below we formulated the research suggestions more specifically and proactively. Moreover, we added an explanation on the difficulty of specific research suggestions and in addition we added a practical suggestion for further development.

Recommendations for future research 

This umbrella review provides an overview of the current stand on research on sleep determinants, useful in day-to-day practice of health professionals addressing sleep issues with their patients. To enhance the practical use of the sleep determinants in daily practice, we recommend the presentation of the determinants in a well-structured and summarizing model, providing a subject for further research. 

Since sleep research is a trending topic it reflects the current availability of reviews on the determinants of sleep in natural circumstances but further research is to be expected. Given the broadness of the topic of sleep, we mainly address directions for further research instead of specific recommendations.

Our suggestions for future research are [Page 39, line 955]:

• More systematic reviews or meta-analyses on bedding/mattresses and on cognitive activity would greatly enhance the quality and usability of the overview.

• More effort into clear definitions of relevant sleep parameters.

• Several consecutive days should be considered when looking at sleep in research and practice.

• More research elaborating on the relation between the different sleep determinants and the effect on accompanying sleep parameters.

• More effort should be put into designing high-quality intervention studies on sleep.

---

## [Decision Letter · Decision Letter 2]

25 Oct 2022

Determinants of natural adult sleep: an umbrella review

PONE-D-22-10437R2

Dear Dr. Philippens,

We’re pleased to inform you that your manuscript has been judged scientifically suitable for publication and will be formally accepted for publication once it meets all outstanding technical requirements.

Kind regards,

Alessandro Silvani, M.D., Ph.D.

Academic Editor

PLOS ONE

Additional Editor Comments (optional):

Thank you for your effective and constructive interaction with the Reviewers during the revision of your manuscript.

Please note that in the supplementary file PRISMA_2020_abstract_checklist_umbrella.sleep.docx, line numbers appear incorrect and may refer to an earlier version of the manuscript. Please double-check and/or edit that supplementary file before publication. 

Reviewers' comments:

Reviewer's Responses to Questions

**Comments to the Author**

1. If the authors have adequately addressed your comments raised in a previous round of review and you feel that this manuscript is now acceptable for publication, you may indicate that here to bypass the “Comments to the Author” section, enter your conflict of interest statement in the “Confidential to Editor” section, and submit your "Accept" recommendation.

Reviewer #1: All comments have been addressed

Reviewer #2: All comments have been addressed

Reviewer #3: All comments have been addressed

Reviewer #4: All comments have been addressed

2. Is the manuscript technically sound, and do the data support the conclusions?

Reviewer #1: Yes

Reviewer #2: Yes

Reviewer #3: Yes

Reviewer #4: Yes

3. Has the statistical analysis been performed appropriately and rigorously? 

Reviewer #1: Yes

Reviewer #2: Yes

Reviewer #3: N/A

Reviewer #4: N/A

4. Have the authors made all data underlying the findings in their manuscript fully available?

Reviewer #1: Yes

Reviewer #2: Yes

Reviewer #3: Yes

Reviewer #4: Yes

5. Is the manuscript presented in an intelligible fashion and written in standard English?

Reviewer #1: Yes

Reviewer #2: Yes

Reviewer #3: Yes

Reviewer #4: Yes

6. Review Comments to the Author

Reviewer #1: I re-read with interest the exceptionally thorough revision made by the authors to their paper “Determinants of natural adult sleep: an umbrella review”. From my part the authors have addressed all the requested points.

Reviewer #2: (No Response)

Reviewer #3: Again, as in the previous revision, the main issues raised by Reviewers have been addressed. The manuscript is further improved.

Reviewer #4: The authors adequately addressed the main points raised by providing a general framework for the interpretation of studies reviewed and a more proactive research agenda.

I have no further comments

7. PLOS authors have the option to publish the peer review history of their article (what does this mean?). If published, this will include your full peer review and any attached files.

Reviewer #1: **Yes: **Luca Baldelli

Reviewer #2: No

Reviewer #3: No

Reviewer #4: **Yes: **Marco Filardi
